# Extracellular matrix signatures of human mammary carcinoma identify novel metastasis promoters

Alexandra Naba[1,2]*, Karl R Clauser[3], John M Lamar[1], Steven A Carr[3], Richard O Hynes[1,2]*

[1]David H Koch Institute for Integrative Cancer Research, Massachusetts Institute of Technology, Cambridge, United States; [2]Howard Hughes Medical Institute, Massachusetts Institute of Technology, Cambridge, United States; [3]Proteomics Platform, Broad Institute of MIT and Harvard, Cambridge, United States

**Abstract** The extracellular matrix (ECM) is a major component of tumors and a significant contributor to cancer progression. In this study, we use proteomics to investigate the ECM of human mammary carcinoma xenografts and show that primary tumors of differing metastatic potential differ in ECM composition. Both tumor cells and stromal cells contribute to the tumor matrix and tumors of differing metastatic ability differ in both tumor- and stroma-derived ECM components. We define ECM signatures of poorly and highly metastatic mammary carcinomas and these signatures reveal up-regulation of signaling pathways including TGFβ and VEGF. We further demonstrate that several proteins characteristic of highly metastatic tumors (LTBP3, SNED1, EGLN1, and S100A2) play causal roles in metastasis, albeit at different steps. Finally we show that high expression of LTBP3 and SNED1 correlates with poor outcome for ER⁻/PR⁻breast cancer patients. This study thus identifies novel biomarkers that may serve as prognostic and diagnostic tools.

*For correspondence: anaba@
mit.edu (AN); rohynes@mit.edu
(ROH)

**Competing interests:** The authors declare that no competing interests exist.

**Reviewing editor**: Elaine Fuchs, Rockefeller University, United States

## Introduction

With an estimated number of new cases in 2008 of 1.3 million worldwide (http://globocan.iarc.fr), breast cancer is the second most frequent cancer worldwide and claimed the lives of over 39,000 patients in the United States in 2012 alone. Most cancer deaths (~90%) are due to the metastatic colonization of distant organs by the tumor cells. The development of novel diagnostic approaches, including novel screening and imaging methods to improve detection of smaller primary tumors and distant metastases, and the identification of novel prognostic markers to permit better classification of breast cancers according to their metastatic potential is imperative as is the development of novel therapeutic strategies.

For the last decade, there has been increasing interest in the tumor microenvironment, which encompasses all the components (cellular and acellular) of a tumor in addition to the tumor cells themselves (*Joyce and Pollard, 2009*; *Egeblad et al., 2010*). Indeed, in order to progress, a tumor needs to be surrounded by a permissive environment and, to metastasize, tumor cells need to find or create a favorable niche (seed and soil theory) (*Paget, 1889*; *Ribatti et al., 2006*; *Erler and Weaver, 2009*; *Comen, 2012*). To create such a niche, tumor cells either directly alter the microenvironment, or instruct local or recruited stromal cells to do so. In the context of breast cancer, recent studies have shown that the cross-talk and interaction of the tumor cells with their surrounding microenvironment is necessary for tumor progression (*Tlsty and Coussens, 2006*; *Polyak et al., 2009*; *Dvorak et al., 2011*; *Boudreau et al., 2012*; *Conklin and Keely, 2012*; *Fordyce et al., 2012*). In addition, the nature of the tumor microenvironment or gene expression profile of the stromal cells has been used to define

**eLife digest** Metastasis is the process whereby tumor cells spread within the body and is the cause of most deaths from cancer. This complex process involves several steps: first the cancer cells invade the tissues that surround the tumor; second, the cancer cells enter the blood stream and travel throughout the body; and third, the cancer cells seed the growth of new tumors in distant organs.

Within tissues, the extracellular matrix forms a complex scaffold of proteins that surrounds cells, to support and organize them: it also provides signals that control how much cells can multiply, how likely cells are to stick together or migrate, and even a cell's chances of survival. Pathologists have used an accumulation of extracellular matrix proteins in tumors as a sign that the outcome of the disease will likely be unfavorable for a patient, and that treatment will be challenging. However, we still do not have a clear picture of the composition of the tumor extracellular matrix and we do not know all the details of how it affects tumor growth and metastasis.

Now, Naba et al. have explored these questions by injecting different types of human breast tumor cells into mice. Some of the cells were capable of spreading throughout the body and were said to have a high 'metastatic potential'; others were less capable of spreading and were said to have a low metastatic potential. Naba et al. then analyzed the proteins that made up the extracellular matrix of the tumors that grew in the mice. Some proteins were found in both types of tumor; whereas some proteins were only found in the tumors with low metastatic potential and some were only found in the highly metastatic tumors. Naba et al. also demonstrated that both cancer cells and non-cancer cells—which are also found within the tumors—contributed to the production of the extracellular matrix in the tumor. Moreover, and somewhat surprisingly, the contributions from the non-cancer cells in the two types of tumors were also different.

Computational analysis predicted that the production of several extracellular matrix proteins in the highly metastatic tumors was under the control of signaling pathways that are involved in cancer progression. Furthermore, Naba et al. also demonstrated that several of the extracellular matrix proteins specific to highly metastatic tumors were required for the cancer to spread. These proteins are involved in different stages of the metastatic process, and some of them are commonly over-produced in tumors from patients with some of the worst chances of recovery.

If similar results are consistently observed in clinical samples from humans, the work of Naba et al. could help doctors to discriminate between tumors that will spread and those that will not, which should lead to improved patient care. The proteins and pathways associated with the highly metastatic tumors could be also investigated as potential drug targets.

human breast cancer types (*Bergamaschi et al., 2008*; *Finak et al., 2008*; *Conklin et al., 2011*). Recent studies have also demonstrated that treatment outcome depends on the tumor microenvironment (vascularization, oxygenation, recruited normal cells, etc; *Chauhan et al., 2011*; *Jacobetz et al., 2013*).

The extracellular matrix (ECM) is the complex scaffold of proteins that provides the architectural support for cell and tissue organization (*Hynes and Naba, 2012*). Cells are in turn able to adhere to the extracellular matrix via different types of receptors including the integrins (*Hynes, 2002*; *Geiger and Yamada, 2011*). In addition to providing biophysical cues, the ECM provides biochemical signals that are major regulators of cell proliferation, survival, migration, and invasion. (*Hynes, 2009*). Pathologists have used excessive ECM deposition (desmoplasia) as a marker of tumors with poor prognosis long before the complexity of the ECM was even deciphered (*Anastassiades and Pryce, 1974*). Despite its great importance in physiological (development, aging) and pathological processes such as cancer, the extracellular matrix remains underexplored (*Wilson, 2010*).

To define the composition of extracellular matrices of tissues and tumors, we have developed a proteomics-based method to enrich and identify ECM proteins and coupled it with a bioinformatic annotation of the 'matrisome' defined as the ensemble of ECM and ECM-associated proteins (*Naba et al., 2012*). Using this approach, we characterized the extracellular matrices of normal murine tissues (e.g., lung and colon) and demonstrated that each of these comprises over 100 proteins. In this study, we apply this proteomics approach to study the composition of the ECMs of poorly and highly metastatic human mammary carcinoma xenografts, and show that both the tumor cells and the stromal cells contribute in characteristic ways to the production of the tumor ECM. Moreover, we show that

both tumor- and stroma-derived proteins differ between tumors of different metastatic potential. Importantly, we demonstrate functional roles for several specific tumor-cell-derived ECM proteins in promoting metastasis. Altogether, our results demonstrate that the proteomic analysis of the tumor ECM can be used to identify proteins playing causal roles in tumor progression that could be further developed as prognostic or diagnostic markers and potentially as therapeutic targets.

## Results

### The composition of the tumor extracellular matrix changes with tumor metastatic potential

To identify ECM proteins important for breast cancer progression and metastasis formation, we used a xenograft model where human breast cancer cells were orthotopically injected into the mouse mammary fat pad. We used cell lines of differing metastatic potential. The poorly metastatic MDA-MB-231 cell line was established from cells isolated from a pleural effusion sample from a triple-negative breast cancer patient (*Cailleau et al., 1978*). The highly metastatic MDA-MB-231-LM2 line (denoted LM2), was previously selected and characterized for increased metastatic potential to the lungs (*Minn et al., 2005*). 6.5 weeks post-injection, the primary tumors were harvested, ECM proteins were enriched from tumors using the subcellular fractionation protocol described previously (*Naba et al. 2012*), and the composition of the ECM-enriched fractions obtained was characterized by mass spectrometry (*Figure 1A*).

We define the matrisome of a tumor as the ensemble of proteins detected in two independent biological replicates and by at least two peptides in one of the two replicates. According to this definition, the matrisome of MDA-MB-231 tumors is composed of 144 proteins and the matrisome of LM2 tumors is composed of 161 proteins (*Figure 1B*, *Figure 2*, *Figure 2—source data 1*). Comparison of the matrisomes of MDA-MB-231 and LM2 tumors identified 118 proteins expressed by both tumor types. In addition, we detected 26 proteins specific to the poorly metastatic tumors and 43 proteins specific to highly metastatic tumors (*Figure 1B*, *Figure 2*, *Figure 2—source data 1*). According to our previous bioinformatic definition of the matrisome (*Naba et al., 2012*), we further categorized these ECM proteins into two categories: core matrisome proteins and matrisome-associated proteins. The core matrisome comprises ECM glycoproteins, collagens, and proteoglycans. Matrisome-associated proteins include ECM-affiliated proteins, ECM regulators (ECM remodeling enzymes and their regulators), and ECM-associated secreted factors (*Figure 2*, *Figure 2—source data 1*). One can note that the most abundant proteins for each of these categories were detected in both tumor types. Examples of these proteins are fibronectin (FN1), fibrinogen, laminins, HSPG2 or perlecan, collagens I, III, IV, V and VI, and transglutaminase 2 (TGM2). In fact, we have previously reported the expression of these proteins in several normal tissues (*Naba et al., 2012*), which suggests that these proteins are widely expressed. Most of the matrisome-associated components were detected at lower abundance, reflecting their presence at lower molar ratios than structural core ECM proteins (*Figure 2*, *Figure 2—source data 1*). Therefore, the analysis identified ECM signatures of poorly (26 proteins) or highly (43 proteins) metastatic tumors (*Figures 1B and 2*) and the proteins that differ between the two tumor types belong mostly to the ECM glycoproteins and the ECM regulators categories. Among the 43 proteins characteristic of highly metastatic tumors (*Figure 3D*), we identified several proteins that have previously been reported to play a role in breast cancer metastasis in diverse cell and tumor models. Examples of these are angiopoietin-like 4 (ANGPTL4, *Padua et al., 2008*), cathepsin B (CTSB, *Sevenich et al., 2010*, *2011*; *Vasiljeva et al., 2006*), insulin-like growth factor-binding protein 4 (IGFBP4, *Ryan et al., 2009*) and lysyl-oxidase-like 2 (LOXL2, *Barry-Hamilton et al., 2010*). Several other proteins in our list have been shown to play roles in metastasis of other tumor types ('Discussion').

### Both the tumor- and stroma-derived ECM proteins differ with the tumor metastatic potential

A strength of human/mouse xenograft model systems coupled to mass spectrometry is that they allow the distinction of human (tumor-derived) protein sequences from their murine (stroma-derived) counterparts (*Naba et al., 2012*). Therefore we could determine the relative contributions of tumor and stromal cells to the production of the tumor ECM. We found that the tumor extracellular matrix is secreted by both tumor cells and stromal cells; some proteins are exclusively secreted by the tumor cells or the stromal cells; and some proteins are secreted by both compartments: either in equal

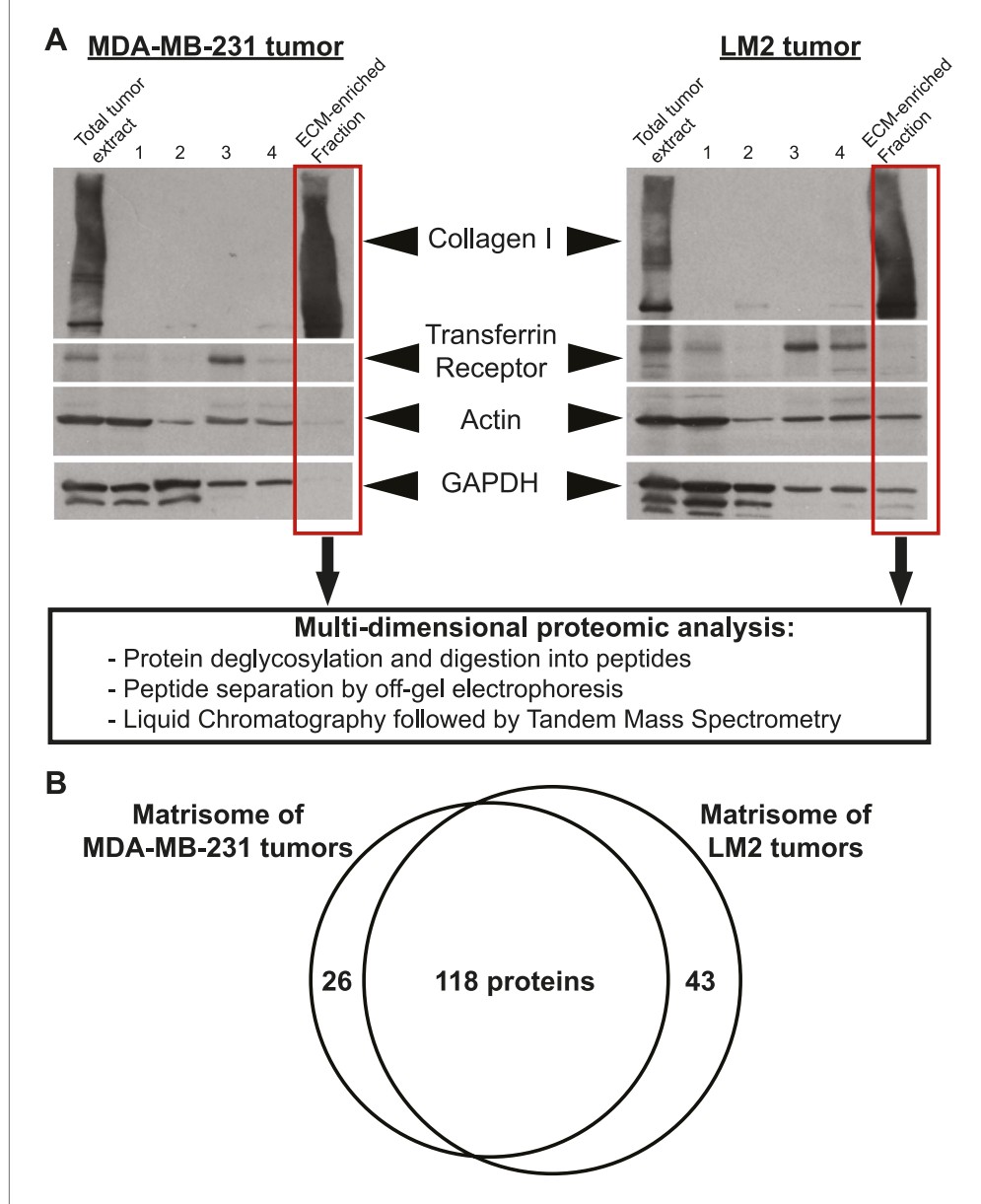

**Figure 1**. Enrichment of extracellular matrix proteins from human mammary tumor xenografts. (**A**) The sequential extraction of intracellular components was monitored by immunoblotting for GAPDH (cytosol), the transferrin receptor (plasma membrane), and actin (cytoskeleton). The remaining insoluble fraction was highly enriched for ECM proteins (collagen I panel) and largely depleted for intracellular components. The ECM-enriched fraction obtained is subsequently submitted to multidimensional proteomic analysis and the matrisome (ECM composition) of each tumor type is defined as the ensemble of proteins present in two replicate samples and with at least two peptides in one of the two replicates. (**B**) Venn diagram represents the comparison of the matrisomes of MDA-MB-231 and LM2 tumors. In addition to 118 ECM proteins detected in both tumor types, we identified 26 proteins specific to poorly metastatic (MDA-MB-231) tumors and 43 proteins characteristic of highly metastatic tumors (LM2).

proportions or more abundantly by the tumor cells or by the stromal cells (*Figure 3*, *Figure 3—source data 1*). We chose a conservative significance threshold of fivefold change in protein abundance, estimated as the summed precursor-ion intensity of the species-specific peptides. Because the tumor–stroma distinction relies on measurement of similar peptides of differing sequence (with accompanying mass differences), it is impractical to use more accurate mass spectrometric relative quantitation approaches that incorporate isotopic labels.

**Core Matrisome — ECM Glycoproteins**

| Entrez Gene Symbol | MW (kDa) | MDA-MB-231 | LM2 |
|---|---|---|---|
| Fn1 | 266.5 | 127 | 145 |
| Fbn1 | 332.9 | 87 | 107 |
| Tnc | 237.5 | 81 | 93 |
| TGFBI | 75.3 | 48 | 40 |
| Fgb | 55.4 | 44 | 46 |
| Postn | 90.9 | 43 | 26 |
| Fga | 61.8 | 31 | 34 |
| Fgg | 51.0 | 31 | 29 |
| Lamb2 | 203.7 | 28 | 19 |
| LAMA5/A3 | 412.3 | 22 | 34 |
| Nid1/Nid2 | 139.5 | 22 | 22 |
| THBS1 | 133.4 | 22 | 19 |
| Lamc1 | 182.9 | 20 | 15 |
| Emilin1 | 108.9 | 20 | 11 |
| PXDN | 167.9 | 17 | 30 |
| ECM1 | 65.2 | 15 | 27 |
| LTBP1 | 195.2 | 15 | 26 |
| ELN | 69.2 | 13 | 14 |
| Lamb1-1 | 209.7 | 13 | 14 |
| LAMB3 | 133.5 | 11 | 14 |
| Vtn | 55.6 | 10 | 12 |
| IGFBP7 | 30.2 | 8 | 12 |
| Fbln2 | 132.2 | 8 | 6 |
| Lama4 | 204.5 | 7 | 9 |
| Dpt | 24.6 | 6 | 2 |
| Efemp1 | 57.2 | 5 | 10 |
| LTBP4 | 182.6 | 4 | 10 |
| Vwa5a | 87.9 | 4 | 7 |
| THSD4 | 115.9 | 4 | 5 |
| IGFBP3 | 32.7 | 4 | 4 |
| LAMC2 | 134.9 | 4 | 4 |
| MFAP2 | 21.6 | 4 | 3 |
| LTBP2 | 204.2 | 3 | 18 |
| CYR61 | 44.2 | 3 | 6 |
| Mfap5 | 19.0 | 3 | 4 |
| GAS6 | 81.7 | 2 | 2 |
| Tnxb | 454.3 | 23 | - |
| Emilin2 | 118.6 | 7 | - |
| Lama2 | 351.9 | 5 | |
| Thbs2 | 133.8 | 3 | |
| FBLN5 | 52.5 | 3 | |
| Hmcn1 | 608.1 | 3 | |
| CTGF | 40.3 | - | 8 |
| Vwf | 322.6 | - | 7 |
| TINAGL1 | 53.8 | - | 6 |
| Vwa1 | 44.8 | - | 3 |
| Papln | 145.4 | - | 3 |
| EMID2 | 46.1 | - | 2 |
| SNED1 | 158.3 | | 8 |
| LTBP3 | 146.5 | | 6 |
| AGRN | 223.0 | | 6 |
| SRPX | 52.8 | | 5 |
| IGFBP4 | 29.1 | | 2 |
| EFEMP2 | 51.8 | | 2 |
| MFGE8 | 43.9 | | 2 |

**Core Matrisome — Collagens**

| Entrez Gene Symbol | MW (kDa) | MDA-MB-231 | LM2 |
|---|---|---|---|
| Col6a3 | 288.4 | 144 | 123 |
| Col1a1/2a1 | 139.1 | 129 | 150 |
| Col12a1 | 341.4 | 120 | 114 |
| Col3a1 | 140.2 | 104 | 104 |
| Col1a2 | 130.1 | 95 | 116 |
| Col5a2 | 146.0 | 43 | 37 |
| Col6a1 | 109.6 | 41 | 37 |
| Col4a2 | 168.5 | 37 | 33 |
| Col5a1/11a1-a2 | 184.4 | 35 | 28 |
| COL7A1 | 296.2 | 29 | 44 |
| Col6a2 | 111.5 | 29 | 23 |
| Col4a1/a3/a5 | 161.8 | 26 | 22 |
| Col5a3 | 172.7 | 23 | 20 |
| Col14a1 | 194.2 | 17 | 4 |
| COL18A1 | 154.4 | 11 | 22 |
| Col16a1 | 157.6 | 9 | 7 |
| Col10a1 | 66.8 | 7 | 17 |
| Col15a1 | 140.9 | 7 | 5 |
| COL13A1 | 70.4 | 6 | 8 |
| COL4A4 | 165.9 | 6 | 3 |
| COL19A1 | 116.0 | 8 | |
| Col6a6/Gm7455 | 248.1 | 7 | |
| COL17A1 | 150.8 | 3 | |
| Col28a1 | 119.7 | 2 | |
| COL22A1 | 162.2 | - | 4 |
| COL24A1 | 176.3 | | 7 |
| Col4a6 | 165.3 | | 6 |

**Core Matrisome — Proteoglycans**

| Entrez Gene Symbol | MW (kDa) | MDA-MB-231 | LM2 |
|---|---|---|---|
| Hspg2 | 479.7 | 80 | 86 |
| Bgn | 42.1 | 12 | 9 |
| Dcn | 40.2 | 6 | 5 |
| Aspn | 43.0 | 8 | - |
| Lum | 38.7 | 6 | - |
| Prelp | 43.6 | 9 | - |
| OGN | 34.3 | 3 | |
| PRG4 | 152.3 | | 3 |

**Matrisome-associated — ECM-affiliated Proteins**

| Entrez Gene Symbol | MW (kDa) | MDA-MB-231 | LM2 |
|---|---|---|---|
| ANXA1 | 38.9 | 29 | 41 |
| ANXA6 | 76.2 | 28 | 25 |
| ANXA2 | 40.7 | 22 | 24 |
| ANXA3 | 36.5 | 15 | 15 |
| LGALS1 | 15.1 | 12 | 10 |
| ANXA5 | 36.0 | 11 | 10 |
| LMAN1 | 57.8 | 8 | 11 |
| ANXA11 | 54.7 | 8 | 11 |
| LGALS3 | 26.2 | 7 | 5 |
| ANXA7 | 53.0 | 5 | 7 |
| Lgals9 | 40.4 | 3 | 5 |
| LGALS8 | 40.4 | 2 | 5 |
| Cspg4 | 253.1 | 2 | 2 |
| C1qa | 26.3 | | 3 |
| Plxnb2 | 209.1 | | 3 |
| C1qc | 26.4 | | 2 |

**Matrisome-associated — Secreted Factors**

| Entrez Gene Symbol | MW (kDa) | MDA-MB-231 | LM2 |
|---|---|---|---|
| HCFC1 | 210.7 | 15 | 18 |
| S100A6 | 10.2 | 6 | 5 |
| S100A11 | 11.9 | 5 | 4 |
| S100A9 | 13.3 | 5 | 3 |
| S100A4 | 12.0 | 4 | 4 |
| CRLF3 | 50.3 | 3 | 5 |
| WNT16 | 25.0 | 2 | 3 |
| FLG2 | 249.4 | 3 | |
| S100A16 | 11.9 | 2 | |
| S100A2 | 11.3 | - | 2 |
| S100a8 | 10.4 | - | 1 |
| ANGPTL4 | 45.6 | | 6 |
| Il16 | 143.1 | | 4 |
| HCFC2 | 87.7 | | 3 |
| S100A10 | 11.3 | | 2 |

**Matrisome-associated — ECM Regulators**

| Entrez Gene Symbol | MW (kDa) | MDA-MB-231 | LM2 |
|---|---|---|---|
| TGM2 | 78.5 | 45 | 48 |
| Plg | 93.5 | 24 | 34 |
| Itih1 | 101.5 | 21 | 5 |
| PLOD1 | 84.1 | 20 | 13 |
| SERPINH1 | 46.6 | 19 | 23 |
| Pzp | 167.2 | 18 | 11 |
| F2 | 71.7 | 14 | 26 |
| F13a1 | 83.7 | 12 | 8 |
| P4HA1 | 61.3 | 11 | 19 |
| PLOD2 | 87.8 | 11 | 7 |
| SERPINB9 | 43.0 | 11 | 7 |
| ADAMTSL1 | 199.4 | 9 | 17 |
| SERPIND1 | 57.2 | 9 | 10 |
| PLOD3 | 85.4 | 8 | 6 |
| MMP14 | 66.2 | 7 | 10 |
| LEPRE1 | 91.8 | 7 | 8 |
| Serpinf2 | 55.2 | 7 | 7 |
| Itih2 | 106.3 | 7 | 6 |
| Loxl1 | 67.2 | 7 | 4 |
| Kng1 | 54.3 | 7 | 3 |
| Serpina3k | 47.1 | 6 | 9 |
| P4HA2 | 61.3 | 6 | 6 |
| SERPINB6 | 43.0 | 6 | 4 |
| PLAU | 45.7 | 5 | 7 |
| Bmp1/Tll1/Tll2 | 113.9 | 4 | 9 |
| CTSD | 45.1 | 4 | 7 |
| Serping1 | 55.9 | 3 | 6 |
| Hrg | 60.1 | 3 | 4 |
| LOX | 47.6 | 2 | 6 |
| CSTB | 11.2 | 2 | 3 |
| NGLY1 | 75.5 | 4 | - |
| MMP1 | 54.2 | 3 | - |
| MMP19 | 57.6 | 2 | - |
| Timp3 | 24.9 | 2 | - |
| Itih3 | 99.7 | 8 | |
| PLAT | 65.1 | 2 | |
| F10 | 56.7 | - | 8 |
| Itih4 | 104.8 | - | 7 |
| Serpinc1 | 52.5 | - | 5 |
| EGLN1 | 46.9 | - | 3 |
| F13b | 78.4 | - | 3 |
| Serpine2 | 44.3 | - | 3 |
| CTSB | 38.8 | | 9 |
| LOXL2 | 88.8 | | 7 |
| CTSC | 52.6 | | 7 |
| ADAM10 | 86.2 | | 6 |
| TIMP1 | 23.9 | | 5 |
| Habp2 | 64.4 | | 5 |
| Serpina1b | 46.1 | | 5 |
| P4HTM | 63.7 | | 3 |
| ADAM9 | 93.1 | | 3 |
| HTRA1 | 52.2 | | 3 |
| CTSF | 54.0 | | 3 |
| LEPREL2 | 82.6 | | 3 |
| CST3 | 16.0 | | 2 |

Legend:
Number of peptides: Low to High
No peptides detected
- Detected in only one of the 2 replicate samples

**Figure 2**. Comparison of the matrisomes of MDA-MB-231 tumors and LM2 tumors identifies ECM proteins characteristic of poorly and highly metastatic tumors. Color code represents the number of unique peptides for each protein from poorly metastatic (MDA-MB-231) or highly metastatic (LM2) human mammary tumors. Values used to generate the figure were extracted from *Figure 2—source data 1*, columns P and AA (number of peptides). Grayed
*Figure 2. Continued on next page*

*Figure 2. Continued*

cells indicate that no peptides were detected in either of the two replicate samples. A dash (–) indicates that the protein was detected in only one of the two replicate samples of a given tumor type or with only one peptide in both replicate samples.

The following source data and figure supplements are available for figure 2:

**Source data 1**. Complete MS data set of ECM-enriched fraction from MDA-MB-231 and LM2 human mammary tumor xenografts.

**Figure supplement 1**. Validation of the differential expression of proteins identified by proteomics.

The majority of proteins (82) found in the matrisomes of poorly and highly metastatic tumors are secreted by the same compartment in both tumor types (*Figure 3A*). As expected, ECM and ECM-associated proteins involved in hemostasis and found in the circulation are murine (fibrinogen chains: α, β, γ, plasminogen, thrombin [Factor 2], Factor XIII transglutaminase, vitronectin, etc). Structural collagens (collagens I and V) are also mostly secreted by the stroma. Components of the basement membranes were found to be expressed by the tumor cells only (laminin chains α3, β3, γ2), by both compartments (laminin chains α5, γ1, COL6A3, HSPG2) or by the stroma (Nidogens 1 and 2). In addition, 36 proteins were detected in both tumor types but their origin differed as a function of the tumor's metastatic potential (*Figure 3B*). In poorly metastatic tumors, fibronectin (FN1) and periostin (POSTN) are secreted by the tumor cells, whereas in highly metastatic tumors, they are secreted by both the tumor cells and the stromal cells. Conversely, several basement membrane components (laminin chains β1 and β2, and the collagen chains 4A1, 4A5 6A1, 6A2 and 18A1) are secreted in poorly metastatic tumors mostly by the stroma, whereas in highly metastatic tumors, they are secreted by both the tumor cells and the stromal cells. As expected, many proteins specific to either poorly or highly metastatic tumors were secreted exclusively by the tumor cells. However, interestingly, we also observed that the stromal compartment also contributed to the differences detected (*Figure 3C,D*, *Figure 3—source data 1*). This indicates that tumor cells of differing metastatic potential not only synthesize distinct subsets of ECM proteins but also influence which ECM proteins are produced by the stroma.

## Probing the tumor extracellular matrix reveals the activation of major signaling cascades

We sought to determine whether the expression of the ECM proteins characteristic of highly metastatic mammary tumors was controlled by common upstream regulators and whether, by probing the tumor ECM, we could identify signaling pathways contributing to tumor progression. To do so, we used Ingenuity Pathway Analysis (*Figure 4—figure supplement 1*). We found that 17 of the 43 (nearly 40%) ECM proteins found to be up-regulated in LM2 tumors are downstream of the TGFβ signaling pathway (*Figure 4*, left panel). The majority of these are tumor-derived, although several host proteins (C1qa, C1qc and Vwf) are also identified as being downstream of TGFβ. Interestingly, the LM2 cell line and its metastatic potential have previously been shown to be dependent on TGFβ (*Minn et al., 2007*).

One of the other signaling pathways significantly represented within our data set is the HIF1α/VEGF pathway that influences the levels of 10 of the 43 ECM proteins identified in our study (*Figure 4*, right panel). We also noted that there is a large overlap between the targets of TGFβ and HIF1α/VEGF as the genes identified as being downstream of the HIF1α/VEGF pathway are all also downstream of the TGFβ pathway. Interestingly, a recent publication by Curran and Keely highlights the convergence of TGFβ and HIF1α signaling pathways as a key contributor to the cross-talk between breast tumor cells and stromal cells (*Curran and Keely, 2013*). Altogether, this analysis indicates that by interrogating the tumor extracellular matrix, one can identify signaling pathways up-regulated within the tumor cells or stromal cells and these represent potential targets for therapeutic intervention.

## Tumor-cell-derived ECM proteins influence the metastatic dissemination of tumor cells to distant organs

As noted above, several of the proteins up-regulated in LM2 tumors have previously been implicated in metastasis. We next wanted to evaluate whether other proteins found to be differentially expressed between poorly and highly metastatic tumors might play any causal, functional roles in tumor progression. Accordingly, we selected a set of proteins (LTBP3—Latent TGFβ Binding Protein 3;

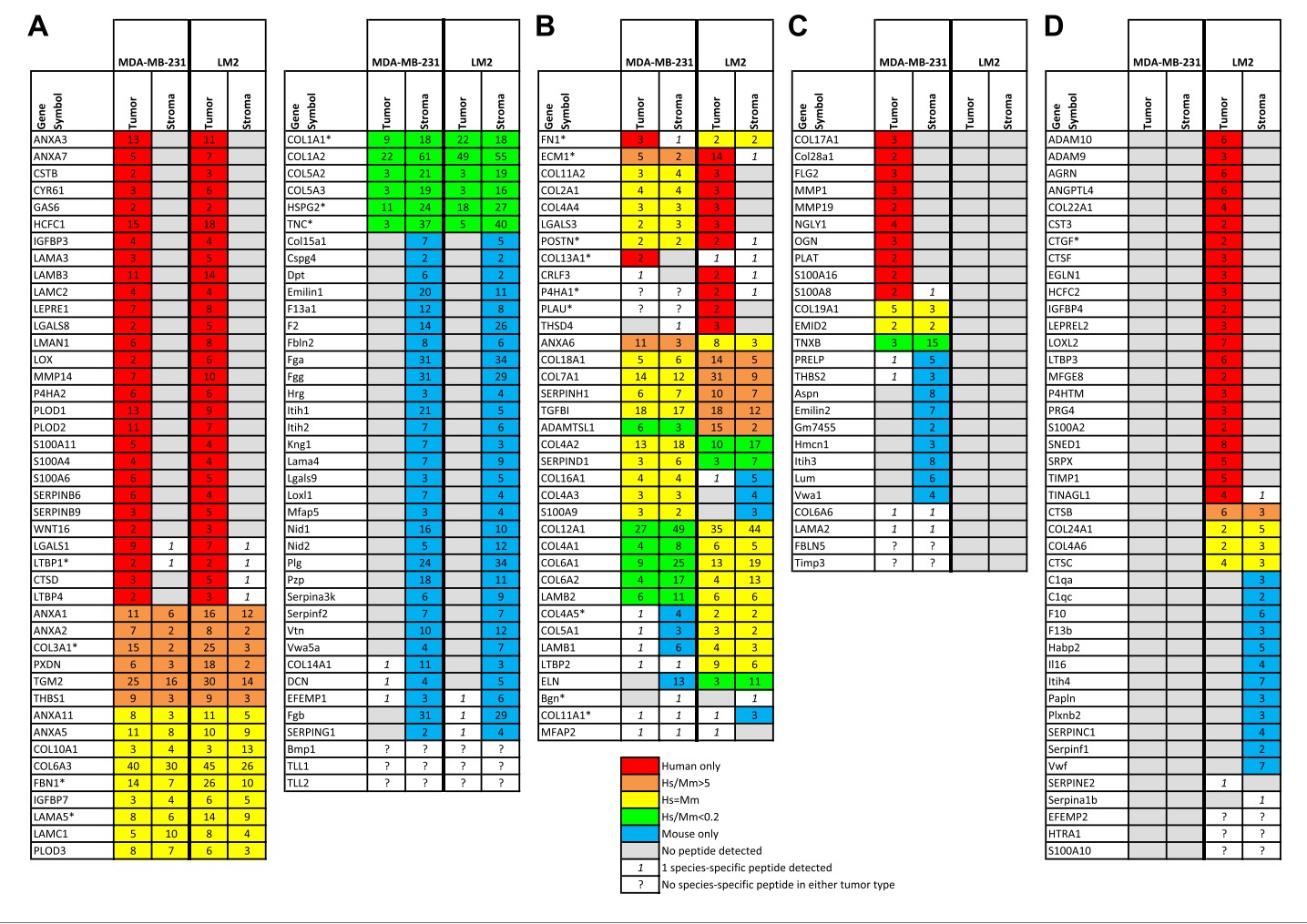

**Figure 3**. The tumor extracellular matrix is secreted by both tumor cells and stromal cells and differs with the tumor's metastatic potential. (**A**) Proteins expressed by both tumor types and by the same compartment in the two tumor types. (**B**) Proteins expressed by both tumor types but by different compartments. (**C**) Proteins secreted by MDA-MB-231 tumors and not by LM2 tumors. (**D**) Proteins secreted by LM2 tumors and not by MDA-MB-231 tumors. The number of peptides detected for each protein is indicated. For the proteins secreted by both the tumor cells and the stromal cells, the number of peptides listed corresponds to the number of human (tumor-derived)- or murine (stroma-derived)-specific peptides. For the proteins secreted by only one compartment, the number of peptides includes both species-specific and indistinguishable peptides. Proteins are sorted by tumor type and by their origins: tumor (red), stroma (blue), or both (yellow: similar abundance of the human and mouse proteins, orange: human form is at least five times more abundant than the mouse form, green: the mouse form is at least five times more abundant). To determine the relative contributions of the tumor and stromal cells to the secretion of ECM proteins, human-to-mouse peptide abundance ratios were calculated using the values indicated in column P and AB for the MDA-MB-231 and LM2 tumors respectively (**Figure 3—source data 1**). Proteins for which different isoforms have been detected are indicated with an asterisk (*) and, for simplicity, isoforms are combined here, their UniProt accession numbers can be found in **Figure 3—source data 1**, column AP. In a few instances, the origin of the protein could not be determined due to the lack of species-specific peptides; these proteins are indicated with a question mark (?).

The following source data are available for figure 3:

**Source data 1**. Complete MS data set of ECM-enriched fraction from MDA-MB-231 and LM2 human mammary tumor xenografts taking into account the origin of matrisome proteins.

**Source data 2**. Detailed list of all of the confidently identified peptide spectrum matches (PSMs) from the LC-MS/MS runs of the 11 fractions resulting from off-gel electrophoresis of each of the two MDA-MB-231 tumor replicates and each of the two LM2 tumor replicates.

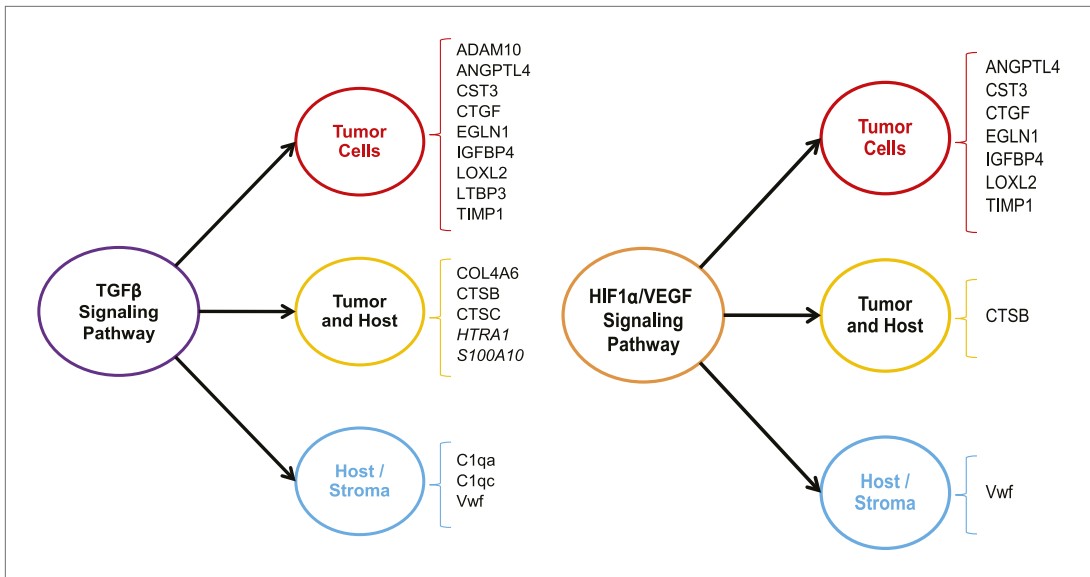

**Figure 4**. The TGFβ and the HIF1α/VEGF pathways are up-regulated in highly metastatic mammary tumors. The 43 ECM and ECM-associated proteins (tumor- and/or stroma-derived) unique to highly metastatic mammary tumors were uploaded to the Ingenuity Pathway Analysis software and queried for common upstream regulators. The analysis revealed an enrichment of TGFβ and HIF1α/VEGF targets within our ECM signature of highly metastatic mammary tumors.

The following figure supplements are available for figure 4:

**Figure supplement 1**. Ingenuity Pathway Analysis.

SNED1—Sushi, Nidogen and EGF-like Domains 1; EGLN1—Egl Nine homolog 1 and S100A2) based on the following criteria: they were (1) detected only in highly metastatic tumors; (2) secreted exclusively by the tumor cells; (3) in relatively high abundance (*Figures 2 and 3D*, *Figure 2—source data 1*, *Figure 3—source data 1*); (4) not previously investigated for any roles in metastasis. We focused on ECM proteins produced by the tumor cells as their expression can easily be manipulated in vitro. We also investigated the role of CYR61, a tumor-derived ECM protein detected in both MDA-MB-231 and LM2 tumors but with a much greater abundance in LM2 tumors (*Figures 2 and 3*, *Figure 2—source data 1*, *Figure 3—source data 1*). For proteins against which antibodies were available (LTBP3, EGLN1, and CYR61) we confirmed their differential expression by immunoblot and/or immunofluorescence analyses (*Figure 2—figure supplement 1A,B*). We also performed RT-qPCR analyses for several other proteins in our lists despite the fact that it is well known that mRNA levels and steady-state protein levels do not correlate at all well (*Gygi et al., 1999*). It can be seen that the mRNA changes for several of the proteins of interest (CYR61, EGLN1, LTBP3) do not show mRNA changes corresponding with the marked differences in protein levels detected, although some do (S100A2, SNED1) (*Figure 2—figure supplement 1C*). In addition, we show that Ltbp3 and Sned1 mRNAs are up-regulated during malignant progression in the autochthonous MMTV-PyMT murine mammary tumor model (*Figure 2—figure supplement 1D*).

To address the roles of these proteins in tumor progression, we knocked down each of the five proteins selected for analysis in LM2 cells using short hairpins. As control, we used an shRNA targeting the firefly luciferase (sh-Cont.). For each gene of interest, two distinct hairpins giving efficient knockdown were selected (*Figure 5—figure supplement 1*). Control or knocked-down LM2 cells were orthotopically injected into the mouse mammary fat pad. At sacrifice (7 +/−0.5 weeks post-injection), the masses of the primary tumors were measured. We observed that none of the genes, when knocked down, significantly affected primary tumor growth (*Figure 5A*). Consistently, we also observed no differences in tumor cell proliferation or apoptosis between control and knockdown tumors (*Figure 5—figure supplement 3*). Importantly, the primary tumors (and the metastases deriving from them) were

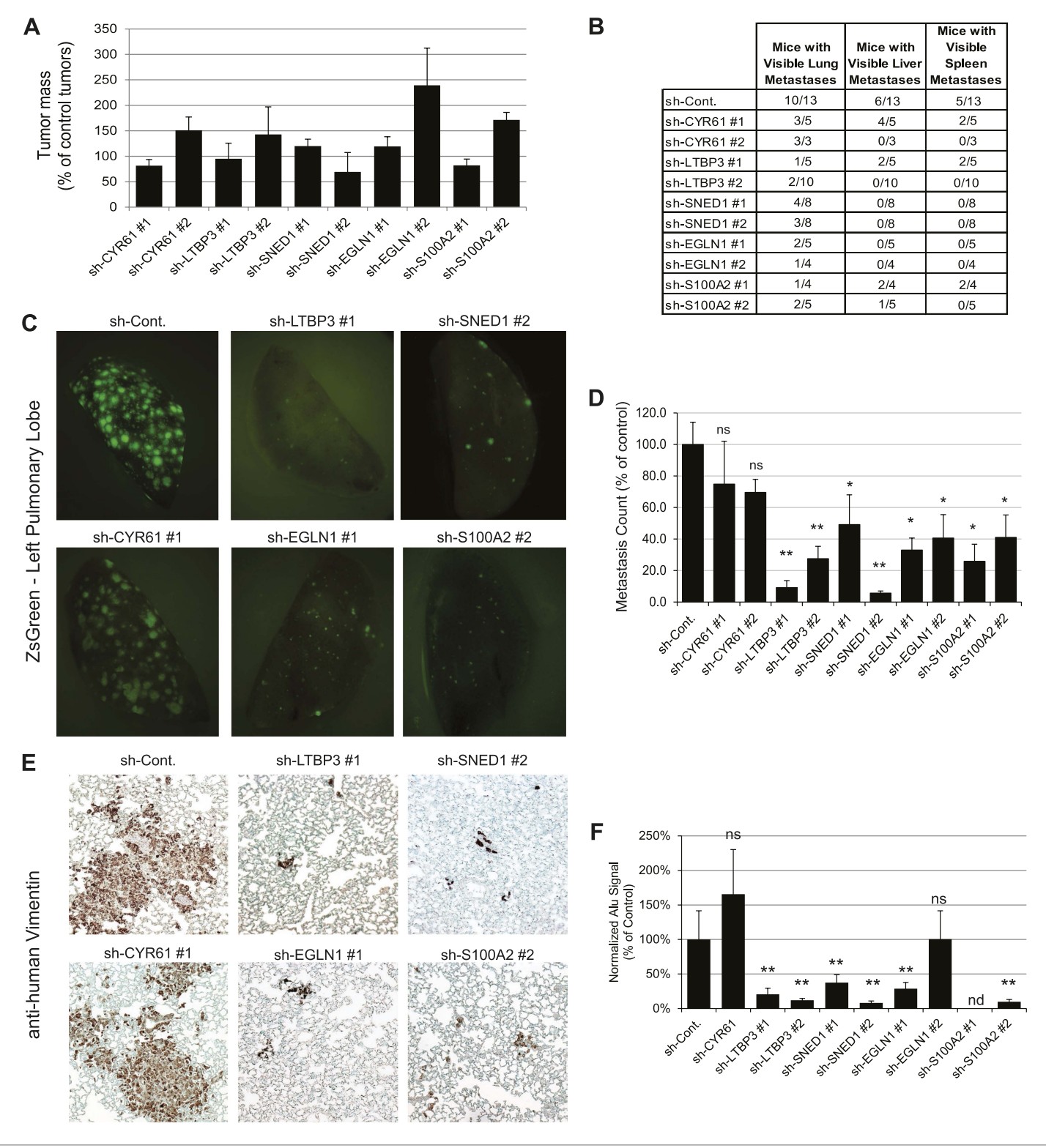

**Figure 5**. Tumor-cell-derived ECM proteins influence the metastatic dissemination of tumor cells to distant organs. Mice were injected orthotopically with control or knockdown LM2 cells. Tumors were allowed to grow for 7 ± 0.5 weeks. Number of mice per condition is indicated in *Figure 5B*. (**A**) ECM protein knockdown does not inhibit primary tumor growth. At sacrifice, control and knockdown primary tumors were weighed. Bar chart represents the mass of knockdown tumors as a percentage of that of control tumors ± SEM. Student's *t* test was performed and none of the genes affected significantly and consistently primary tumor growth. (**B**) LM2 control tumors metastasize to the lungs, liver and spleen. The number of mice that presented with visible

*Figure 5. Continued on next page*

*Figure 5. Continued*

metastases in the indicated organs is indicated. (**C**) Representative pictures of whole left pulmonary lobe from LM2 (control or knockdown)-tumor-bearing mice with ZsGreen-positive metastatic foci. (**D**) Numbers of ZsGreen-positive metastatic foci in the left pulmonary lobe were counted. Data are presented as percentage of control ± SEM (Student's *t* test, *p<0.05, and **p<0.01). Number of animals per group is indicated in *Figure 5B*. (**E**) Lung sections were stained with a human-specific anti-vimentin antibody to detect human tumor cells in the murine lung. (**F**) Alu PCR was performed on genomic DNA extracted from the lungs of control or knockdown tumor-bearing mice. Data are presented as normalized Alu signal as compared to murine actin signal and as percentage of control ± SEM (Student's *t* test, *p<0.05, **p<0.01, ns: not significant, and nd: not determined). Number of animals per group is indicated in *Figure 5B*.

The following figure supplements are available for figure 5:

**Figure supplement 1**. Establishment of stable LM2 knockdown cells.

**Figure supplement 2**. Persistence of gene expression knockdown in tumors.

**Figure supplement 3**. Tumor-cell-derived ECM proteins do not influence proliferation or apoptosis of primary mammary tumors.

still ZsGreen-positive, indicating that the shRNA construct was still strongly expressed (*Figure 5C*). We also confirmed that the knockdowns persisted in vivo (*Figure 5—figure supplement 2*) and that there was no compensatory up-regulation of these genes by the stroma (data not shown). These data indicate that none of these genes is required for primary tumor growth.

LM2 cells have been selected for their high metastatic potential and their tropism for the lungs (*Minn et al., 2005*). In our system, LM2 cells are injected in the absence of exogenous ECM support (Matrigel) in immunodeficient mice (NOD/SCID/IL2Rγ-null). We observed that the parental LM2 cells and the LM2 cells expressing the control hairpin, in addition to metastasizing to the lung, also could colonize the liver and the spleen from the primary site (*Figure 5B*). We thus wanted to evaluate the capacity of the knockdown tumors to metastasize from the primary site (mammary gland) to distant organs. Whereas knocking down CYR61, a protein that we detected by proteomics in both poorly and highly metastatic tumors, did not affect metastasis formation, knocking down any of the four genes characteristic of highly metastatic (LM2) tumors, and not detected in the poorly metastatic (MDA-MB-231) tumors, led to a significant decrease of the metastatic burden as compared with that of mice injected with control LM2 cells (*Figure 5B*). None of the SNED1 or EGLN1 knockdown tumors disseminated to organs other than the lungs and the number of mice presenting visible lung metastases was dramatically decreased (*Figure 5B*). LTBP3 and S100A2 knockdown also led to a very significant inhibition of metastatic dissemination, although we detected occasional visible metastases in the liver or spleen of mice injected with LTBP3 or S100A2 knockdown cells (*Figure 5B*). Because of the pulmonary tropism of LM2 cells, we further analyzed the lungs of control or knockdown tumor-bearing mice. The analysis of the lungs of control or knockdown tumor-bearing mice for the presence or absence of ZsGreen-positive metastatic foci confirmed the significant decrease in the number of metastases formed by knockdown tumors (*Figure 5B–D*). As the tumor cells are human, we monitored the presence of human cells within murine lungs. We used a human-specific anti-vimentin antibody and Alu PCR to detect human genomic DNA and both methods confirmed the significantly decreased metastatic burden in knockdown-tumor-bearing mice as compared to control mice (*Figure 5E,F*). These results demonstrate that these four ECM proteins play significant functional roles in the metastatic dissemination of these mammary tumor cells.

## Tumor cell-derived ECM proteins influence different steps of the metastatic cascade

Metastatic dissemination is a multistep process that requires tumor cells to proliferate, invade the surrounding host tissue, intravasate, survive in the circulation, extravasate and eventually seed and proliferate in distant organs. To pinpoint which steps of the metastatic cascade are influenced by each metastasis-associated ECM protein, we first compared the vascularization, invasiveness, and degree of fibrosis in control and knockdown primary tumors. Knock-down of any of the five ECM genes selected did not affect significantly primary tumor vascularization (*Figure 6—figure supplement 1*). Control tumors markedly invaded the skin (*Figure 6*) and the surrounding muscles (not shown), and were

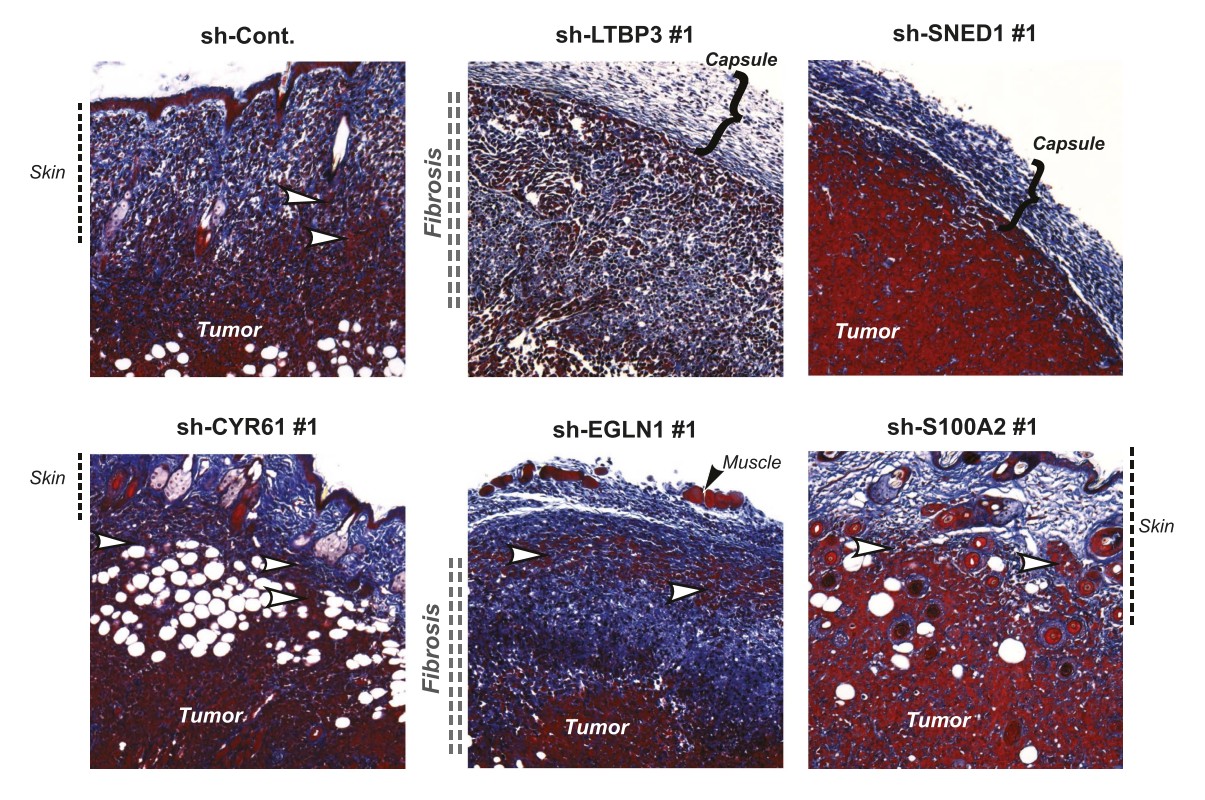

**Figure 6**. Tumor-cell-derived ECM proteins influence the invasiveness of primary mammary tumors. Primary tumor sections were stained with Masson's trichrome (blue: collagen fibers, red: cells) to evaluate fibrosis (indicated by vertical double-dashed line) and encapsulation (bracket) or invasiveness (white arrowhead) into the skin of the primary tumors. Note that tumors in which LTBP3 or SNED1 are knocked down are less invasive and more encapsulated than in the control tumors. CYR61, EGLN1, or S100A2 knockdown did not affect tumors' invasiveness.

The following figure supplements are available for figure 6:

**Figure supplement 1**. Tumor-cell-derived ECM proteins do not influence vascularization of primary mammary tumors.

significantly fibrotic, as indicated by staining tumor sections with Masson's Trichrome (*Figure 6*). Cells in which CYR61, EGLN1, or S100A2 were knocked down formed highly invasive (white arrow) and desmoplastic tumors resembling the control tumors, suggesting that these proteins did not influence tumor invasion or fibrosis (*Figure 6*). Interestingly, LTBP3 and SNED1 knockdown tumors were characterized by deposition of collagen at the periphery of the tumor forming a thick capsule. Consistently, the LTBP3 and SNED1 tumors failed to invade the surrounding tissues, suggesting that LTBP3 and SNED1 are promoters of tumor invasiveness.

To test whether each metastasis-associated ECM protein influenced later steps of the metastatic cascade (survival in the circulation, extravasation and seeding, survival and proliferation in the lung), we injected the tumor cells directly into the circulation via the lateral tail vein. When injected into the blood circulation, control LM2 cells efficiently seeded and colonized the lungs and formed large metastases detectable by fluorescence (*Figure 7A*, upper panel), by staining lung sections with hematoxylin and eosin (*Figure 7A*, middle panel) or with an anti-human vimentin antibody (*Figure 7A*, lower panel). Knockdown of EGLN1 or S100A2 strongly inhibited the formation of pulmonary metastases (*Figure 7A–7C*). In contrast, knockdown of LTBP3 or SNED1 in LM2 cells did not affect lung colonization (*Figure 7A,B*), consistent with our finding that these proteins influence tumor cell invasion at the primary site (*Figure 6*). Although LTBP3 knockdown gave rise to the same number of metastases, the metastases were significantly smaller (based on the size of the ZsGreen-positive foci) and consistently, the Alu PCR signal was significantly lower than in the control lungs (*Figure 7C*). Together, our data suggest that, although all these four proteins are required to increase metastasis formation, they do

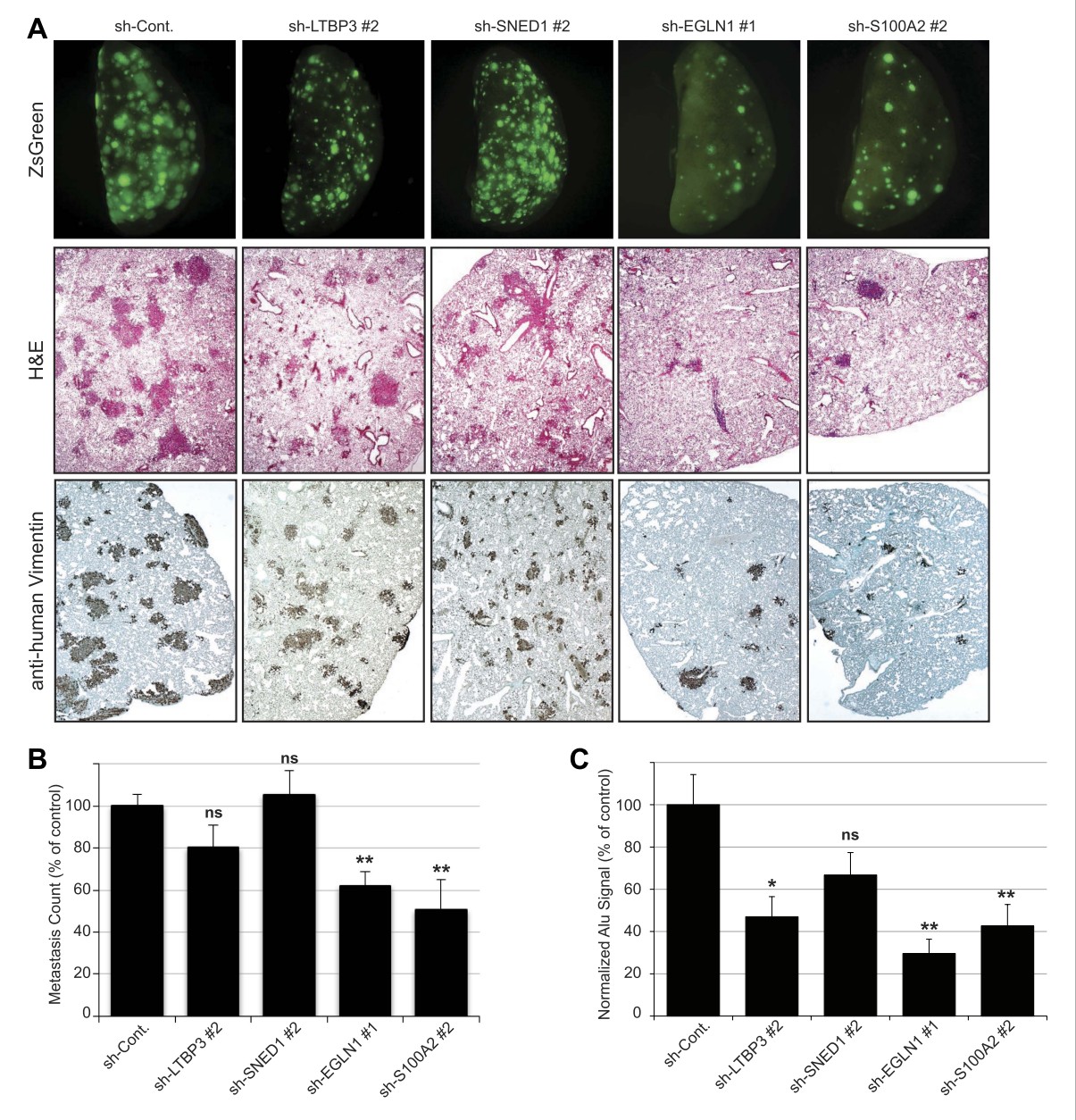

**Figure 7**. Tail vein metastasis assay. Control or knockdown cells were injected via the tail vein and the formation of lung metastases was evaluated. (**A**) Upper panel: representative pictures of the whole left pulmonary lobe with ZsGreen-positive metastatic foci from mice injected with LM2 (control or knockdown) cells (upper panel). Lung sections were stained with hematoxylin and eosin (H&E, middle panel) or with human-specific anti-vimentin antibody (lower panel) to visualize the metastatic foci. (**B**) Numbers of ZsGreen-positive metastatic foci in the left pulmonary lobe. Data are presented as percentage of control ± SEM (Student's *t* test, *p<0.05, **p<0.01, ns: not significant). Number of animals per group: sh-Cont.: 17 mice, sh-LTBP3: 10 mice, sh-SNED1: 10 mice, sh-EGLN1: 8 mice, sh-S100A2: 8 mice. (**C**) Alu PCR was performed to monitor the presence of human tumor cells in the murine lung. Data are presented as percentage of control ± SEM (Student's *t* test, *p<0.05, **p<0.01, ns: not significant). Number of animals per group: sh-Cont.: 17 mice, sh-LTBP3: 10 mice, sh-SNED1: 10 mice, sh-EGLN1: 8 mice, sh-S100A2: 8 mice.

so by influencing different steps of the metastatic cascade. LTBP3 and SNED1 likely act early in the metastatic cascade by promoting tumor invasiveness. In addition, LTBP3, but not SNED1, appears to affect the growth of lung metastases. Our data also suggest that EGLN1 and S100A2 are likely to act at later stages of the metastatic cascade (extravasation, seeding, survival, and/or growth of the tumor cells in secondary sites) because, although the primary tumors are very invasive, they failed to colonize distant organs.

## Tumor-derived ECM proteins may have prognostic value for some breast cancer patients

In addition to providing novel insights into the biology of breast cancer progression, we sought initial indications that any of the ECM proteins identified here by proteomics and shown to play causal roles in mammary carcinoma progression, might correlate with clinical outcome in breast cancer patients. Since equivalent proteomic data are not yet available for human patient material, we could only make comparisons with mRNA expression data, even though we know that correlation between protein and mRNA levels is known to be loose. We used Kaplan–Meier Plotter (*Györffy et al., 2010*), an online-available meta-analysis tool for biomarker assessment. Using this tool, we tested whether the mRNA expression of any of the four genes we studied correlated with distant-metastasis-free survival for breast cancer patients. While expression levels for SNED1 and LTBP3 were not significantly correlated across all mammary cancer sub-types (*Figure 8*, right panels), they were significantly correlated (p values of 0.0079 and 0.034, respectively) with poor prognosis for estrogen-receptor-negative and progester-one-receptor-negative (ER⁻/PR⁻) patients (*Figure 8*, left panels). In this context, it is significant that the MDA-MB-231 cells were isolated from the pleural effusion of a triple-negative breast cancer patient (*Cailleau et al., 1978*). While we are not suggesting that the correlations between the expression of two ECM proteins and probability of metastasis in a limited subset of cancer patients is currently of

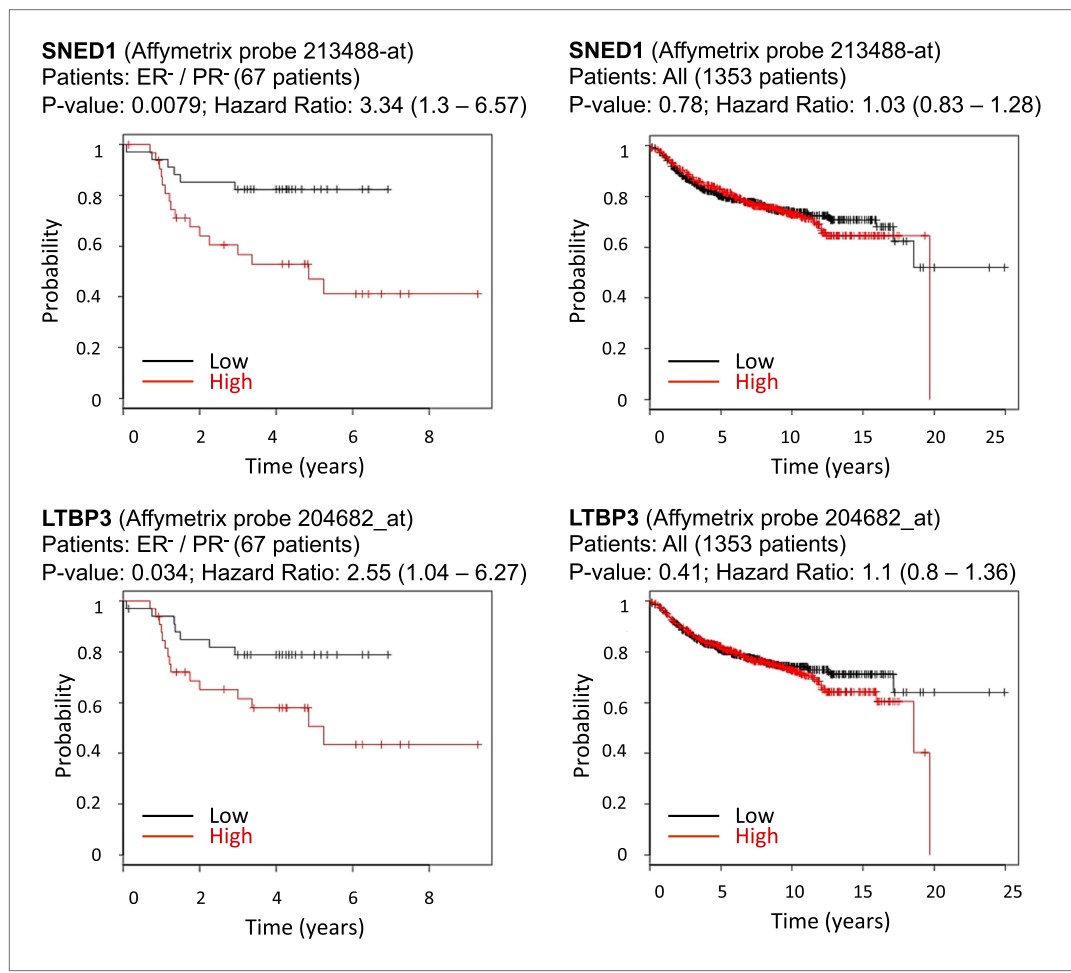

**Figure 8**. Correlation between SNED1 and LTBP3 expression and distant-metastasis-free survival in breast cancer patients. We tested the predictive value of the four ECM genes studied using the online assessment tool Kaplan–Meier Plotter. Whereas the expression of none of the genes studied correlated with distant metastasis-free survival for the entire population of breast cancer patients, SNED1 and LTBP3 expression inversely correlated with the survival of estrogen-receptor-negative and progesterone-receptor-negative (ER⁻/PR⁻) breast cancer patients.

predictive value, this analysis demonstrates that we can identify within our proteomics signatures, a subset of ECM proteins that are pro-metastatic in model systems and may have potential as prognostic biomarkers and are worthy of further investigation.

## Discussion

We demonstrate in this study that a proteomics-based discovery approach can define ECM signatures of tumors of differing metastatic potential. We show that the tumor ECM is derived from both the tumor and stromal cells and that tumors of differing metastatic potential differ in both the tumor- and stroma-derived ECM. Our study led to the identification of tumor-derived ECM proteins that play functional roles in tumor progression and metastatic dissemination. Indeed, a high proportion of the proteins we detected as up-regulated in highly metastatic tumors (some reported previously, some new to this study) contribute to metastatic dissemination. We further demonstrate that these proteins can affect different steps of the metastatic cascade: LTBP3 and SNED1 appear to promote early stages of invasion and dissemination, whereas EGLN1 and S100A2 act at later stages of the metastatic cascade. Finally, using gene expression data sets from breast cancer patients, we show that some of the proteins discovered by proteomics as differentially expressed are correlated with breast cancer patient outcomes. We studied in detail five ECM or ECM-associated proteins not previously clearly implicated in metastasis and demonstrate that four out of five of these proteins play roles at different steps of the metastatic cascade.

CYR61 was detected in both poorly and highly metastatic tumors but with a greater abundance in highly metastatic tumors. Previous studies had reported the up-regulation of CYR61 in more aggressive tumor cell lines or lesions (*Tsai et al., 2000*, *2002*). Our study demonstrates that although produced in greater abundance by highly metastatic tumors, CYR61 is not necessary to promote metastatic dissemination as knocking down its expression did not have any effect on the metastatic dissemination of LM2 cells.

Consistent with the role of TGFβ in cancer progression (*Padua and Massagué, 2009*; *Massague, 2012*), we show that LTBP3, a protein that regulates TGFβ secretion and bioavailability (*Chen et al., 2002*; *Munger and Sheppard, 2011*) promotes primary tumor invasion and metastasis. LTBP3 is also a protein that contributes to the formation of the fibrillar ECM network (*Ramirez and Rifkin, 2009*; *Todorovic and Rifkin, 2012*). Thus, whether LTBP3 contributes to tumor progression because of its signaling role or its architectural role remains to be elucidated. Of note, LTBP3 is a member of a family of proteins and the three other members of the family; LTBP1, LTBP2 and LTBP4 were all detected in the matrisome of both poorly and highly metastatic tumors, LTBP1 and 4 being expressed by the tumor cells and not the stromal cells and LTBP2 being expressed by both the tumor cells and stromal cells (*Figures 2 and 3*, *Figure 2—source data 1*, *Figure 3—source data 1*).

SNED1 (Sushi, Nidogen, and EGF-like Domains 1) is a large ECM glycoprotein first identified as a potential component of the basement membrane (*Leimeister et al., 2004*) whose roles in normal physiology and pathology remain to be characterized. In this study, we report for the first time a functional role for SNED1 in tumor progression and metastasis and we now wish to understand the molecular mechanisms by which SNED1 contributes to tumor progression. Interestingly, SNED1 displays two potential integrin-binding sites, an RGD motif is found upstream of the amino-terminal NIDO domain and an LDV motif is upstream of the first EGF domain.

EGLN1 (also known as PHD2, Prolyl Hydroxylase Domain-containing protein 2) is a prolyl-hydroxylase and, in normoxic conditions, catalyzes the hydroxylation of proline residues of the hypoxia-inducible factor 1 α (HIF1α), rendering it susceptible to poly-ubiquitinylation and subsequent proteasomal degradation (*Berra et al., 2003*). Klotsche-van Ameln et al. have reported that inhibiting EGLN1 had an anti-tumor effect by enhancing TGFβ anti-proliferative effect in a model of murine osteosarcoma (*Klotzsche-von Ameln et al., 2011*). However, the extracellular role of this protein remains unknown.

S100A2 is a calcium-binding protein that belongs to the S100 family of acute-phase response factors and, when overexpressed in non-small-cell lung carcinoma cells subsequently injected subcutaneously in mice, has been shown to promote metastasis to the lungs (*Bulk et al., 2009*). The fact that we observed a decrease in the number of metastases formed by cells in which S100A2 was knockeddown, both when the cells were injected orthotopically and metastasized from the primary tumor and when the cells were injected in the circulation, suggests that S100A2 is important for the growth of tumor cells in distant organs, in particular the lungs.

The proteomics-based discovery pipeline we developed identified, with high yield, proteins playing a causal role in tumor progression and metastatic dissemination. Indeed, in addition to the four proteins whose involvement in mammary carcinoma metastasis we report here, previous studies using various mammary carcinoma cell lines or tumor models had already implicated several other proteins in breast cancer progression (ANGPTL4, CTSB, IGFBP4, LOXL2) or invasiveness of breast tumor cells (ADAM9, ADAM10). In addition, analyses by Kaplan–Meier Plotter (*Győrffy et al., 2010*) show that several proteins from our list of ECM proteins up-regulated in highly metastatic LM2 cells (ADAM9, LOXL2, S100A2; data not shown) show statistically significant correlations with poor prognosis for metastasis-free survival of breast cancer patients. Thus, in total, from the list of 43 ECM proteins that we find expressed only in the metastatic LM2 cells and not in MDA-MB-231, 12 show suggestive evidence of functional involvement in metastasis of breast cancer. Also included are proteins previously implicated in tumor progression in other cancer types (e.g., CTGF, TIMP1, S100A10) and proteins that we have identified in proteomics screens of other tumor models (unpublished data). These concordances indicate that the proteins we identify here are relevant to breast cancer metastasis and their significance is likely not limited solely to the breast cancer lines we used in this study.

To determine whether these proteins are regulated independently or might be coregulated, we conducted pathway analysis. Two main pathways, the TGFβ pathway and the HIF1α/VEGF pathway, appeared to be upstream of several of the proteins we identified. In addition to signal transduction pathways, microRNAs are another mechanism by which cells can rapidly control the expression or inhibition of sets of genes. Korpal et al. have identified in breast cancer patients the miR-200 family as being pro-metastatic via its inhibition of Sec23a, a protein involved in the secretion machinery (*Korpal et al., 2011*). Interestingly, several of the proteins we found to be up-regulated in highly metastatic tumors (AGRN, LTBP3, EFEMP2, IGFBP4, SERPINE2, SNED1 and TINAGL1) were shown in the Korpal study to be down-regulated in the conditioned medium of Sec23a knockdown cells and were thus postulated to be anti-metastatic. However, our study indicates that this set of proteins is pro-metastatic, which could point towards tumor-specific or context-dependent mechanisms of action for these proteins.

Our study also revealed that not only the production of tumor-derived ECM proteins changes with a tumor's metastatic potential but the stromal contribution to the tumor ECM also changes. This is consistent with the idea that the composition of the stroma (possibly including the cell types present) of a non-metastatic primary tumor differs from that of a highly metastatic primary tumor and that tumor cells of different metastatic potential may recruit different stromal cell types. In fact, we have observed that LM2 tumors are more vascularized than the MDA-MB-231 tumors (data not shown). Our data also suggest that tumor cells of different metastatic potential can instruct local stromal cells to express different sets of ECM proteins. This interesting aspect of tumor–stroma interactions has not been extensively studied in vivo, although a recent paper notes that cancer-associated fibroblasts from different mammary tumor subtypes differ in their expression profiles (*Tchou et al., 2012*).

Finally, we believe that our approach offers the possibility of identifying proteins of clinical relevance. We showed that LTBP3 and SNED1, identified by proteomics in our study as differentially expressed between poorly and highly metastatic mammary tumors, correlate at the transcript level with clinical outcome in a cohort of ER⁻/PR⁻ breast cancer patient. We can envision three ways in which one could exploit the ECM to improve cancer diagnosis and prognosis and improve cancer patient care. As demonstrated for SNED1 and LTBP3, other proteins that are part of the proteomic signatures of highly and poorly metastatic breast tumors described here, may prove to correlate with patient outcomes and may prove useful as novel prognostic and diagnostic biomarkers. Development of immuno- or mass spectrometry-based assays (*Gillette and Carr, 2013*) for these proteins is therefore of interest since protein-level measurements may be of equivalent or greater predictive value than RNA expression analyses. This will require development of panels of specific antibodies against the ECM proteins of interest and their testing on larger numbers of well annotated patient samples to establish correlations with tumor stage and patient outcomes. ECM proteins are particularly favorable candidate biomarkers since they are abundant, are laid down in characteristic patterns and are readily accessible. ECM proteins can also serve as anchors to deliver selectively to tumors and/or metastases imaging probes to enhance the detection of metastases or anti-tumor agents to enhance therapy on the model of the work pioneered by Neri et al. (*Pasche and Neri, 2012*). Finally, another possible translational aspect is the identification of potential novel direct therapeutic targets. Whether by targeting the interactions between ECM proteins and their cellular receptors (*Goodman and Picard, 2012*) or by disrupting the architecture of the ECM (that acts as a barrier) to allow more efficient drug

delivery (*Yuan et al., 2007*; *Whatcott et al., 2011*; *Jacobetz et al., 2013*), the proteins discovered in our differential proteomics-based approach have the potential to serve as novel anticancer therapeutic targets.

## Materials and methods

### Cells

The human mammary carcinoma cells, MDA-MB-231, and their highly metastatic variant MDA-MB-231_ LM2 (LM2) were a kind gift of Dr Joan Massague (Memorial Sloan Kettering Cancer Center, New York, NY). The cells were grown in HyClone high-glucose Dulbecco's modified Eagle's medium (Thermo Scientific, Pittsburgh, PA) supplemented with 2 mM glutamine and 10% fetal bovine serum (Invitrogen, Carlsbad, CA) at 37°C in a 5% $CO_2$ incubator.

### Orthotopic human mammary tumor cell xenografts

Tumor growth and spontaneous metastasis formation were assayed by transplanting tumor cells orthotopically into the mammary fat pad of 6 to 8-week-old female NOD/SCID/IL2Rγ-null mice (Jackson Laboratory, Bar Harbor, ME). The mice were anesthetized by intraperitoneal (IP) injection of 125–250 mg/kg body weight of Avertin (reconstituted in PBS), followed by IP injection of 100 μl of 12 μg/ml Buprenorphine for analgesia. A small incision was made in the right flank to expose the inguinal mammary gland, and $2.5.10^5$ cells in suspension in 25 μl of Hank's Balanced Salt Solution (Invitrogen, Carlsbad, CA) were injected. The mice received three additional IP injections of 100 μl of 12 μg/ml Buprenorphine at 12-hr intervals following the surgery. Surgery and mouse monitoring were performed according to the animal protocol approved by MIT's Department of Comparative Medicine and Committee on Animal Care. Animals were sacrificed 7 ± 0.5 weeks post-injection and the tumors were dissected and weighed, flash frozen and kept at −80°C or fixed in 3.8% formaldehyde, imaged with a fluorescence microscope and subsequently embedded in paraffin and sectioned. In addition, secondary tumor sites (lung, liver, spleen) were collected and fixed in 3.8% formaldehyde for subsequent embedding in paraffin and sectioning.

### Experimental lung metastasis assay

$5.10^4$ cells in 100 μl of Hank's Balanced Salt Solution were injected into the lateral tail vein of 6- to 8-week-old female NOD/SCID/IL2Rγ-null mice. The mice were sacrificed 4 weeks post-injection and lungs were inflated with 3.8% formaldehyde imaged with a fluorescence microscope and subsequently fixed overnight with 3.8% formaldehyde. Samples were kept in 70% ethanol prior to embedding and sectioning.

### Quantification of metastases

ZsGreen-positive foci were counted in the left pulmonary lobe using the open-source software Cell Profiler 2.0 (*Lamprecht et al., 2007*; http://www.cellprofiler.org/) and counts were manually curated when needed. The Cell Profiler pipeline developed for this study is provided as a .cp file (*Supplementary file 1*). Student's *t* test was performed to evaluate the statistical significance of the results.

### Tissue preparation and ECM protein enrichment

Sequential extractions of proteins from frozen tumor samples were performed using the CNMCS (Cytosol/Nucleus/Membrane/Cytoskeleton) compartmental protein extraction kit (Cytomol, Union City, CA) as previously described (*Naba et al. 2012*). This led to the extraction of intracellular soluble proteins and the enrichment of ECM proteins.

### ECM-enriched fraction solubilization and digestion

Two independent biological replicates of each tumor type (MDA-MB-231 or LM2) were analyzed. The ECM-enriched samples from each tumor were subsequently analyzed by mass spectrometry.

100–300 μg ECM-enriched pellets were solubilized and reduced in a solution of 8M urea, 100 mM ammonium bicarbonate pH 8, 10 mM dithiothreitol, and incubated at 37°C for 30 min with continuous vortexing. After cooling to room temperature, cysteines were alkylated by adding iodoacetamide to 25 mM for 30 min. After diluting to 2M urea with 100 mM ammonium bicarbonate pH 8.0, samples were deglycosylated with 1000–2000 units of PNGaseF (New England BioLabs, Ipswich, MA) and incubated at 37°C for 2 hr with continuous vortexing, followed by digestion with Lys-C (Wako Chemicals

USA, Inc., Richmond, VA), at a ratio of 1:100 enzyme:substrate, with for 2 hr. Final digestion was done using trypsin (Sequencing Grade, Promega, Madison, WI), at a ratio of 1:50 enzyme:substrate at 37°C overnight with continuous vortexing, followed by a second aliquot of trypsin, at a ratio of 1:100 enzyme:substrate, and an additional 2 hr of incubation. Solutions that began cloudy upon initial reconstitution were clear after overnight digestion. Samples were acidified and desalted using 10 mg HLB Oasis Cartridges (Waters Corp., Milford, MA) eluted with 60% acetonitrile, 0.1% trifluoroacetic acid (TFA), followed by concentration in a Speed-Vac.

### Peptide fractionation by off-gel electrophoresis

Approximately, 50 μg samples of peptide digest were fractionated using an Agilent 3100 OFFGEL Fractionator (Agilent Technologies, Wilmington, DE) and 13 cm Immobiline Drystrips pH 3–10 (GE Healthcare BioSciences AB, Uppsala, Sweden, 17-6001-14). Fractionation was performed according to the Agilent instruction manual. Briefly, peptides were diluted in IPG buffer, pH 3–10 (GE Healthcare, 17-6000-87), containing 5% glycerol. 150 μl of peptide solution were loaded into each of 12 wells and focused for 20 kV hr with a maximum current of 50 μA and power of 200 mW (24–36 hr). Focused solutions were pipetted out of each well and the wells were re-extracted with 30% acetonitrile/0.1% TFA. Fractions 9 and 10 were combined, yielding 11 total fractions for subsequent LC-MS/MS analysis. Fractions were acidified with TFA, cleaned-up using stage tips, that is, pipette tips packed with reversed-phase membrane disks (Empore C-18 #2215, 3M Corporation, St Paul, MN), eluted with 60% acetonitrile, 0.1% TFA, and then concentrated in a Speed-Vac (*Rappsilber et al., 2003*).

### Mass spectrometry

Tryptic digests were analyzed with an automated nano LC-MS/MS system, consisting of an Agilent 1100 nano-LC system (Agilent Technologies, Wilmington, DE) coupled to either an LTQ-Orbitrap or an LTQ Orbitrap XL Fourier transform mass spectrometer (Thermo Fisher Scientific, San Jose, CA) equipped with a nanoflow ionization source (James A Hill Instrument Services, Arlington, MA). Peptides were eluted from a 10-cm column (Picofrit 75 um ID, New Objectives) packed in-house with ReproSil-Pur C18-AQ 3 μm reversed phase resin (Dr Maisch, Ammerbuch Germany) using either a 120 min gradient at a flow rate of 200 nl/min to yield ~20 s peak widths. Solvent A was 0.1% formic acid and solvent B was 90% acetonitrile/0.1% formic acid. The elution portion of the LC gradient was 3–6% solvent B in 2 min, 6–31% B in 75 min, 31–60% B in 13 min, 60–90% B in 1 min, and held at 90% B for 5 min. Data-dependent LC-MS/MS spectra were acquired in ~3 s cycles; each cycle was of the following form: one full Orbitrap MS scan at 60,000 resolution followed by 8 MS/MS scans in the ion trap on the most abundant precursor ions using an isolation width of 3 m/z. Dynamic exclusion was enabled with a mass width of ± 25 ppm, a repeat count of 1 and an exclusion duration of 45 s. Charge-state screening was enabled along with monoisotopic precursor selection and non-peptide monoisotopic recognition to prevent triggering of MS/MS on precursor ions with unassigned charge or a charge state of 1. Normalized collision energy was set to 30 with an activation Q of 0.25 and activation time of 30 ms.

### Protein identification, quantitation, and distinction between mouse (stroma) and human (tumor) proteins

All MS data was interpreted using the Spectrum Mill software package v4.1 beta (Agilent Technologies, Santa Clara, CA). Similar MS/MS spectra acquired on the same precursor m/z within ± 60 s were merged, MS/MS spectra with precursor charge >4 and poor quality MS/MS spectra, which failed the quality filter by not having a sequence tag length >0 (i.e., minimum of two masses separated by the in-chain mass of an amino acid) were excluded from searching. MS/MS spectra were searched against a UniProt database containing both human (78,369 entries) and mouse (53,448 entries) sequences. All sequences (including isoforms and excluding fragments) were downloaded from the UniProt website on 30 June 2010. To each database a set of common laboratory contaminant proteins (73 entries) was appended. Initial search parameters included: ESI linear ion-trap scoring parameters, trypsin enzyme specificity with a maximum of two missed cleavages, 35% minimum matched peak intensity, ± 20 ppm precursor mass tolerance, ± 0.7 Da product mass tolerance, and carbamidomethylation of cysteines and possible carbamylation of N-termini as fixed/mix modifications. Allowed variable modifications were oxidized methionine, deamidation of asparagine, pyro-glutamic acid modification at N-terminal glutamine, and hydroxylation of proline with a precursor MH + shift range of −18 to 97 Da. Hydroxyproline was only observed in the proteins known to have it (collagen and proteins containing collagen domains; emilins, etc) and only within the expected GXPG sequence motifs. *Figure 3—source data 2*

containing the detailed peptide spectral matches might have some examples not in the expected motif when there is either a proline near the motif for which the spectrum could have had insufficient fragmentation to confidently localize the mass change to a particular residue, or a nearby methionine in the peptide and the spectrum had insufficient fragmentation to localize the mass change to oxidized methionine or hydroxyproline. When the motif nX[ST] occurs in a peptide in *Figure 3—source data 2*, this is likely to indicate a site where N-linked glycosylation was removed by the PNGaseF treatment of the sample. While a lowercase n indicates a gene-encoded asparagine residue detected in aspartic acid form, possible mechanisms of modification such as acid-catalyzed deamidation during sample processing vs enzymatic conversion during deglycosylation cannot be explicitly distinguished. Identities interpreted for individual spectra were automatically designated as confidently assigned using the Spectrum Mill autovalidation module to apply target-decoy-based, false-discovery rate (FDR) scoring threshold criteria via a two-step auto-threshold strategy at the spectral and protein levels. First, peptide mode was set to allow automatic variable range precursor mass filtering with score thresholds optimized to yield a spectral level FDR of 1.6% for each of the precursor charge states 2, 3, and 4 in each LC-MS/MS run. Second, protein mode was applied to further filter all the peptide-level validated spectra combined from the 22 LC-MS/MS runs from both replicates of an experiment using a minimum protein score of 20 and a maximum protein-level FDR of zero. Since the maximum peptide score is 25, the protein-level step filters the results so that each identified protein is comprised of multiple peptides unless a single excellent scoring peptide was the sole match. The above criteria yielded false discovery rates of <1.0% for each sample at the peptide–spectrum match level and <1.2% at the distinct peptide level as estimated by target-decoy-based searches using reversed sequences. In calculating scores at the protein level and reporting the identified proteins, redundancy is addressed in the following manner: the protein score is the sum of the scores of distinct peptides. A distinct peptide is the single highest scoring instance of a peptide detected through an MS/MS spectrum. MS/MS spectra for a particular peptide may have been recorded multiple times, (i.e., as different precursor charge states, isolated from adjacent OGE fractions, modified by deamidation at Asn or oxidation of Met) but are still counted as a single distinct peptide. When a peptide sequence >8 residues long is contained in multiple protein entries in the sequence database, the proteins are grouped together and the highest scoring one and its accession number are reported. In some cases, when the protein sequences are grouped in this manner there are distinct peptides which uniquely represent a lower scoring member of the group (isoforms, family members, and different species i.e., mouse vs human). Each of these instances spawns a subgroup and multiple subgroups are reported and counted towards the total number of proteins and in *Figure 3—source data 1*, they are given related protein subgroup numbers (see column AQ and AR I *Figure 3—source data 1A*, e.g., Tenascin C (TNC), the murine and human forms and are listed as subgroup members 25.1 and 25.2 respectively). Our in silico matrisome list was then used to categorize all of the identified protein subgroups as being ECM-derived or not. The reporting of the number of peptides contributing to each subgroup can be altered by enabling the subgroup-specific option in Spectrum Mill. This was done to report separately the species-specific peptides, the peptides common to both human and mouse, and the total of common and species-specific peptides.

Relative abundances of proteins were determined using either the number of peptides or using the peptide abundance (see legends for *Figures 2 and 3*). When we sought to determine the relative abundance of human-derived and murine-derived proteins, we used extracted ion chromatograms (XIC's) for each peptide precursor ion in the intervening high resolution FT-MS scans of the LC-MS/MS runs. An individual protein's abundance was calculated as the sum of the ion current measured for all quantifiable peptide precursor ions with MS/MS spectra confidently assigned to that protein. Peptides were considered not quantifiable if they were shared across multiple subgroups of a protein or the precursor ions had a poorly defined isotope cluster (i.e., the subgroup-specific and exclude poor isotope quality precursor XIC's filters in Spectrum Mill were enabled). Proteins were considered quantifiable if they were represented in two independent samples and represented by at least two distinct peptides in one of the two samples. The peak area for the XIC of each precursor ion subjected to MS/MS was calculated automatically by the Spectrum Mill software in the intervening high-resolution MS1 scans of the LC-MS/MS runs using narrow windows around each individual member of the isotope cluster. Peak widths in both the time and *m/z* domains were dynamically determined based on MS scan resolution, precursor charge and *m/z*, subject to quality metrics on the relative distribution of the peaks in the isotope cluster vs theoretical. Although the determined protein ratios are generally reliable to within a factor of twofold of the actual ratio, numerous experimental factors contribute to

variability in the determined abundance for a protein. These factors may include incomplete digestion of the protein; widely varying response of individual peptides due to inherent variability in ionization efficiency and interference/suppression by other components eluting at the same time as the peptide of interest, differences in instrument sensitivity over the mass range analyzed, and inadequate sampling of the chromatographic peak between MS/MS scans.

The original mass spectra may be downloaded from MassIVE (http://massive.ucsd.edu) using the identifier: MSV000078535. The data should be accessible in the 'Spectrum' folder at ftp://MSV000078535:a@massive.ucsd.edu.

## Gene knock-down, reverse transcription, and quantitative real-time PCR

97-nucleotide miR30-based shRNAs targeting each gene of interest (*Supplementary file 2*) were designed using the shRNA design software available through the Cold Spring Harbor Laboratory website (http://katahdin.mssm.edu/siRNA/RNAi.cgi?type=shRNA), and cloned into the MSCV-ZSG-2A-Puro-miR30 vector retroviral vector as described previously (*Stern et al., 2008*; *Lamar et al., 2012*). This vector expresses the miR30-based shRNA in the 3'UTR of a single transcript encoding the ZsGreen reporter gene and the puromycin-resistance gene. Retroviruses were packaged in 293FT cells and LM2 cells were transduced as previously described (*Stern et al., 2008*; *Lamar et al., 2012*). For qPCR, RNA was isolated from cell or tumor lysates using RNeasy kit (Qiagen, Germantown, MD) and cDNA was synthesized by reverse transcription using the First-Strand cDNA Synthesis Kit (Promega, Madison, WI). qPCR reactions were performed using Bio-Rad SYBR Green Supermix (Bio-Rad, Hercules, CA) according to the manufacturer's instructions. PCR conditions were 95°C for 10 min, followed by 40 cycles of 95°C for 20 s, 58°C for 30 s, and 72°C for 30 s. qPCR data analysis was performed using Bio-Rad CFX Manager Software. Human and murine PCR primers used are listed in *Supplementary file 3*.

## Immunohistochemistry

Tumor samples were formaldehyde-fixed and paraffin-embedded. Sections were dewaxed and rehydrated following standard procedures. When required, heat-induced epitope retrieval was performed by incubating sections in 10 mM sodium citrate buffer (pH6.0) heated at 95°C for 20 min. Sections were cooled down at room temperature and were then stained using appropriate Vectastain ABC kits (Vector Laboratories, Burlingame, CA). Primary antibodies used were: mouse anti-human vimentin antibody (Leica, Davie, FL), mouse anti-Ki67 antibody (Vector Laboratories), rabbit anti-cleaved caspase 3 (Cell Signaling, Danvers, MA), rabbit anti-CD31 (PECAM) antibody (Abcam, Cambridge, MA), rabbit anti-LTBP3 (Santa Cruz Biotechnology, Dallas, TX), Sections were subsequently counterstained with methyl green or hematoxylin. Hematoxylin and eosin and Masson's trichrome stainings were performed following standard procedures.

## Immunoblotting

Tumor samples were lysed in Laemmli buffer, proteins were separated by SDS-PAGE and immunoblotting was performed using the following antibodies: rabbit anti-collagen I (Millipore, Billerica, MA), mouse anti-transferrin receptor (Invitrogen), mouse anti-GAPDH (Millipore), rabbit anti-LTBP3 (Santa Cruz Biotechnology), rabbit anti-EGLN1 (Cell Signaling), the rabbit anti-actin antibody was generated in the laboratory. The rabbit anti-CYR61 antibody was a kind gift of Dr Lester Lau.

## Alu PCR

DNA was extracted from formaldehyde-fixed and paraffin-embedded tissues using the QIAamp DNA FFPE Tissue Kit (Qiagen). Quantitative PCR was conducted on 1 ng of genomic DNA. Primers used are: Alu Forward: 5'GTGAAACCCCGTCTCTACTAAAAATACAAA3', Alu Reverse: 5'GCGATCTCGGCTCACTGCAA3'. qPCR reactions were performed using Bio-Rad SYBR Green Supermix (Bio-Rad) according to the manufacturer's instructions. PCR conditions were 95°C for 10 min, followed by 40 cycles of 95°C for 20 s, 58°C for 30 s, and 72°C for 30 s qPCR data analysis was performed using Bio-Rad CFX Manager Software. Murine actin served as a reference to normalize qPCR data.

## Data mining

Ingenuity Pathway Analysis (http://www.ingenuity.com/products/ipa) was used to identify common upstream regulators ('Results'). Kaplan–Meier Plotter (http://www.kmplot.com/), an online-available meta-analysis tool (*Györffy et al., 2010*), was used to test possible correlations between expression of

the genes encoding proteins identified by proteomics in our study ('Results') with distant metastasis-free survival. We interrogated the data from all patients or only the subset of estrogen-receptor-negative and progesterone-receptor-negative patients included on the multiple clinical data sets available.

## Acknowledgements

The authors wish to thank Laurie Tang, Denise Crowley, Emavieve Coles, and Shahinoor Begum for technical assistance, Roderick Bronson from the Hope Babette Tang Histology Facility at the Swanson Biotechnology Center for assistance with interpretation of histopathology, Mark Bray from the Imaging Platform of the Broad Institute for assistance with Cell Profiler, and the members of the Hynes Lab for helpful discussions.

## Additional information

### Funding

| Funder | Grant reference number | Author |
|---|---|---|
| National Cancer Institute - Tumor Microenvironment Network | U54 CA126515/CA163109 | Richard O Hynes |
| National Cancer Institute - David H. Koch Institute Support Grant | P30-CA14051 | Richard O Hynes |
| Howard Hughes Medical Institute | | Alexandra Naba, Richard O Hynes |
| Broad Institute of MIT and Harvard | | Karl R Clauser, Steven A Carr, Richard O Hynes |
| Ludwig Center for Cancer Research | | Alexandra Naba, Richard O Hynes |
| NIH / National Research and Service Award | | John M Lamar |
| National Cancer Center | | John M Lamar |

The funders had no role in study design, data collection and interpretation, or the decision to submit the work for publication.

### Author contributions

AN, KRC, Conception and design, Acquisition of data, Analysis and interpretation of data, Drafting or revising the article; JML, Acquisition of data, Drafting or revising the article; SAC, Conception and design, Drafting or revising the article; ROH, Conception and design, Analysis and interpretation of data, Drafting or revising the article

### Ethics

Animal experimentation: All work involving mice was approved by the MIT Committee on Animal Care (Protocol # 0412-033-15, Approval Date: 4/19/2012, MIT Animal Welfare Assurance No. A-3125-01). All experiments are covered by approved protocols reviewed by the MIT Committee on Assessment of Biohazards (Protocol # 156, Approval date 3/21/13).

## Additional files

### Supplementary files

• Supplementary file 1. Cell Profiler pipeline to quantify ZsGreen-positive metastatic foci. The Cell Profiler pipeline designed to quantify ZsGreen-positive metastatic foci in whole organs is provided as a .cp file. The file can be opened in any text editor software or using Cell Profiler.

• Supplementary file 2. shRNA sequences.

• Supplementary file 3. qPCR primer sequences.

## Major dataset

The following dataset was generated:

| Author(s) | Year | Dataset title | Dataset ID and/or URL | Database, license, and accessibility information |
|---|---|---|---|---|
| Naba A, Clauser KR, Lamar JM, Carr SA, Hynes RO | 2013 | Proteomics analysis of ECM of poorly and highly metastatic mammary tumor xenografts | MSV000078535; ftp:// MSV000078535:a@ massive.ucsd.edu/ | Publicly available at MassIVE (http:// massive.ucsd.edu/). |

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
