## [Decision Letter]

Thank you for sending your work entitled “Extracellular matrix signatures of human mammary carcinoma identify novel metastasis promoters” for consideration at *eLife*. Your article has been evaluated by a Senior editor and 4 reviewers, one of whom is a member of our Board of Reviewing Editors, and one of whom, Joan Brugge, has agreed to reveal her identity.

All four of your reviewers agree that the work is of high quality and comes to a number of interesting and important conclusions that highlight the importance of ECM proteins in tumor progression. However, we are concerned that the conclusions made about the potential role of these ECM proteins as regulators of metastasis in human breast cancer are not currently well supported by the experiments. The consensus is that there are several major concerns that currently pose a barrier to publishing in *eLife*.

1) One of the greatest and nearly unanimous concerns is the use of a single pair of cultured cell lines (one cell type) of “normal and cancer states” (derived from a single patient). The consensus is that this is simply not enough to judge whether the findings are at physiologically relevant and functionally significant and how broadly applicable the results might be. The cell lines used were manipulated extensively and passaged through mice several times, which further compounds the problems/caveats. A related issue is whether the differences in overall matrisome and in the five ECM proteins tested truly represent differences between mammary tumors and lung metastases, as you hope is the case, or whether they represent ECM differences between tumors that are initiated with 250,000 cells implanted at once versus tumors that are initiated by circulating cells that reach the lungs over time singly or as very small clusters. Since mammary tumors are among the few that can be successfully initiated at the near single cell level, the recommendation to re-visit matrisome results (at least the key differentially expressed proteins identified) on mammary tumors initiated with just a few cells instead of the 250,000.

2) Another broadly shared criticism is that xenograft studies are not sufficient to draw conclusions about human tumors. As pointed out, expression of these ECM proteins in human tumors of different metastatic potential is a necessity to generalize about the relevance to breast cancer. It will also be important to demonstrate that tumors with the same metastatic potential show similar ECM composition.

3) One reviewer raises a concern that most of the proteins that showed differential expression and all of the ones chosen for functional studies had relatively few peptides in the proteomics analyses. Alternative methods should be used to validate expression differences and further delineate their magnitude.

4) Several reviewers raised the concern as to whether some of the expression changes are a consequence rather than a cause of metastasis. This issue needs to be sorted out and substantiated experimentally.

5) Some follow up studies seems warranted to support the conclusions made about the roles of matrisome components not previously implicated in metastasis, but unearthed in your study.

[Editors’ note: the authors asked the editors for clarifications in advance of resubmission, which are shown below.]

We all agree that you and your co-authors have found interesting candidate ECM proteins potentially linked to metastatic potential. All of your reviewers concluded that some independent experimental confirmation of findings would be necessary for publication in *eLife*.

Although you and your co-authors concede that language linking these proteins to human breast cancer metastases should be reduced in the manuscript, it was felt that secondary confirmation of the role of these proteins in (minimally) another breast cancer cell line would be essential in order for the conclusions following from this work to be of broad interest to the readership of *eLife*. For example, you indicate that an antibody for LTBP3 for western blot and immunostaining is available. Perhaps your coauthors could perform additional knock down studies and characterization of this candidate protein and its role in a second breast cancer cell line.

The validation of protein expression levels for the top candidates which your coauthors have already performed should be included in the paper. Indeed, you have clearly recognized the importance of validating your results in primary human tumors but state that this work is beyond the scope of this publication. While it was agreed that a full study of primary tumors would be unnecessary for this publication, without further characterization using a second xenograft model, the most interesting implications of this work are not well supported by the data. I hope this helps to streamline the process and delineate more specifically what would be the minimum needed for *eLife* to move forward with publication of your study.

---

## [Author Response]

*All four of your reviewers agree that the work is of high quality and comes to a number of interesting and important conclusions that highlight the importance of ECM proteins in tumor progression. However, we are concerned that the conclusions made about the potential role of these ECM proteins as regulators of metastasis in human breast cancer are not currently well supported by the experiments. The consensus is that there are several major concerns that currently pose a barrier to publishing in* eLife.

We had not intended to claim that all of these ECM proteins were conclusively identified as “regulators of metastasis in human breast cancer”. On rereading the text, we noted several places where we may have inadvertently given that impression. We have modified those sentences and inserted additional text noting what would need to be done to validate these as useful biomarkers for human clinical breast cancer. We acknowledge that, as with any novel large-scale discovery studies, additional follow-up studies are required to establish definitive human patient relevance. In order to extend the implications to human patient material, we have established collaborations with clinicians to access well annotated human patient samples and we are starting the necessary large-scale production of antibodies against ECM proteins. That will take significant time and effort and is beyond the scope of this paper.

*1) One of the greatest and nearly unanimous concerns is the use of a single pair of cultured cell lines (one cell type) of “normal and cancer states” (derived from a single patient). The consensus is that this is simply not enough to judge whether the findings are at physiologically relevant and functionally significant and how broadly applicable the results might be*.

We used the well studied and widely used MDA-MB-231/LM2 pair of human cell lines as discovery tools to evaluate 1) the composition of the ECM of tumors of different metastatic potentials; 2) the tumor vs stroma/host contributions to the production of the tumor ECM; 3) the functional involvement in metastasis of four ECM proteins previously not known to play any role in that process. Those experiments could not have been done without suitable cell lines and mouse xenograft models. A major advantage of this system is that LM2 cells metastasize from the primary tumor site (mammary gland) without requiring the addition of Matrigel (which would complicate the proteomics). LM2 also efficiently metastasize when injected via the tail vein. We find these cells, previously used in several discovery studies (at the RNA expression level) of metastasis, suitable for the protein-level studies that we conducted. We clearly demonstrate in this paper that the changes in the proteins that we tested are functionally significant in metastasis in the systems analyzed. As noted above, we are not claiming that the results obtained with this pair of cell lines are demonstrably broadly applicable. We hope they will be and we have noted the evidence consistent with that hope. We believe that to repeat the complete set of proteomics experiments and knockdowns presented in this manuscript on other tumor cell lines would not be a productive use of time, money and animals (quite apart from requiring probably at least a year of further work). If we were to do that as requested, the question would still remain – how relevant are these data to human breast cancer? It does not seem to us to be a reasonable request. We believe that the appropriate next step is to analyze patient-derived material and we are beginning those studies now but they will in themselves be a large undertaking beyond the scope of this paper.

*The cell lines used were manipulated extensively and passaged through mice several times, which further compounds the problems/caveats*.

We acknowledge that the cell lines are model systems – albeit ones that have been widely used in the literature to good effect as discovery tools. Note that we identified multiple ECM proteins that have previously been implicated in breast cancer as well as discovering numerous additional proteins that change and we demonstrated that 4/5 tested were indeed pro-metastatic.

*A related issue is whether the differences in overall matrisome and in the five ECM proteins tested truly represent differences between mammary tumors and lung metastases, as you hope is the case, or whether they represent ECM differences between tumors that are initiated with 250,000 cells implanted at once versus tumors that are initiated by circulating cells that reach the lungs over time singly or as very small clusters. Since mammary tumors are among the few that can be successfully initiated at the near single cell level, the recommendation to re-visit matrisome results (at least the key differentially expressed proteins identified) on mammary tumors initiated with just a few cells instead of the 250,000*.

This is a misunderstanding: the proteomic analyses were conducted on ECM of primary tumors of differing metastatic potential. The ECM signatures therefore correlate with metastatic potential not with differences between primaries and metastases – a different but also interesting question (a subject of our ongoing work). We added text to clarify this point.

*2) Another broadly shared criticism is that xenograft studies are not sufficient to draw conclusions about human tumors. As pointed out, expression of these ECM proteins in human tumors of different metastatic potential is a necessity to generalize about the relevance to breast cancer. It will also be important to demonstrate that tumors with the same metastatic potential show similar ECM composition*.

As noted above, we agree and we did not intend to conclude that we have yet shown that these ECM proteins are relevant to clinical cancer. The model systems used in this paper are discovery tools and the xenograft approach is essential to dissect the relative contributions of tumor and stromal cells – a set of results that the referees agree are interesting. It will, of course, be necessary (as we are now doing) to extend the work to human patient material. To do so, it will be essential to analyze a larger number of tumors than is feasible by proteomics. Rather, it will be necessary to prepare antibodies to the candidate ECM biomarkers discovered here and apply them in immunohistochemistry assays of human tumor microarrays to test correlations with clinical stage, tumor type, metastatic potential, etc. This is a large endeavor (few useful antibodies are available) but it is only made possible by the discoveries described in this paper – that is why we believe that this paper merits publication prior to that considerable extension to clinical material.

*3) One reviewer raises a concern that most of the proteins that showed differential expression and all of the ones chosen for functional studies had relatively few peptides in the proteomics analyses. Alternative methods should be used to validate expression differences and further delineate their magnitude*.

It is not entirely clear to us why the reviewer is concerned: certainty of identification, quantitation, or both? The 5 proteins chosen for functional studies LTBP3, SNED1, EGLN1, S100A2, and CYR61 were observed with 6, 8, 3, 2, and 6 peptides, respectively, and in duplicate analyses (see [Supplementary-material SD2-data]). Our requirement of a protein being detected in two replicates and with at least two peptides in one of the two replicate is a conservative threshold that was applied to virtually eliminate the possibility of a false-positive identification. So, there is no doubt about the identity or differential expression of the proteins. Rather than using the words most and all, the reviewer’s comment would seem to be more accurately stated as “some of the proteins that showed differential expression and some of the ones chosen for functional studies had relatively few peptides”. We have modified the text to clarify some of the nuances of our methodology. Note that because the tumor-stroma distinction relies on measurement of similar peptides of differing sequence (with accompanying mass differences), it is impossible to use more accurate mass spectrometric quantitation approaches that incorporate isotopic labels.

While it is impractical to validate by alternative methods the protein expression measurements for all 150+ ECM proteins observed in our proteomics experiments, we have included immunoblot data (for LTBP3, EGLN1 and CYR61) and immunohistochemical staining (LTBP3) that confirm the differential expression of these proteins (see Figure 2—figure supplement 1). We note however that evaluating the relative abundance changes for proteins deposited extracellularly is not as simple as surveying mRNA expression. In fact, RNA expression only poorly correlates with protein level expression. However, we have conducted some such experiments and included some data along those lines (see Figure 2—figure supplement 1 and Results section).

*4) Several reviewers raise the concern as to whether some of the expression changes are a consequence rather than a cause of metastasis. This issue needs to be sorted out and substantiated experimentally*.

We are not sure what is meant by this comment. Some of the protein expression changes may indeed be consequences of the fact that the primary tumors differ in their metastatic potential. The reason why we conducted the knockdown experiments was precisely to test whether any of the proteins were causal and, indeed, 4/5 of those tested were causal – others may not be. However, even if some of the changes detected are consequential rather than causal, these proteins may still be valuable as biomarkers as we now note in the revised manuscript (see Results and Discussion sections).

*5) Some follow up studies seems warranted to support the conclusions made about the roles of matrisome components not previously implicated in metastasis, but unearthed in your study*.

We are not sure what is being suggested here – that is exactly why we conducted the knockdown experiments.

*[Editors’ note: the authors asked the editors for clarifications in advance of resubmission, which are shown*
*below.]*

*We all agree that you and your co-authors have found interesting candidate ECM proteins potentially linked to metastatic potential. All of your reviewers concluded that some independent experimental confirmation of findings would be necessary for publication in* eLife*.*

*Although you and your co-authors concede that language linking these proteins to human breast cancer metastases should be reduced in the manuscript, it was felt that secondary confirmation of the role of these proteins in (minimally) another breast cancer cell line would be essential in order for the conclusions following from this work to be of broad interest to the readership of* eLife*. For example, you indicate that an antibody for LTBP3 for western blot and immunostaining is available. Perhaps your coauthors could perform additional knock down studies and characterization of this candidate protein and its role in a second breast cancer cell line*.

*The validation of protein expression levels for the top candidates which your coauthors have already performed should be included in the paper. Indeed, you have clearly recognized the importance of validating your results in primary human tumors but state that this work is beyond the scope of this publication. While it was agreed that a full study of primary tumors would be unnecessary for this publication, without further characterization using a second xenograft model, the most interesting implications of this work are not well supported by the data. I hope this helps to streamline the process and delineate more specifically what would be the minimum needed for eLife to move forward with publication of your study*.

1) A request for independent experimental confirmation of the findings. We take this to mean, in part, confirmation by orthogonal means of the differences detected and documented extensively by proteomics. We have added data confirming the protein-level differences for proteins on which we focused in our tests for causality. Such confirmation either requires decent antibodies (which are not always available but we have included all that are) or repeat proteomics (which we had already included).

We also understood that you and the referees would like to see qRT-PCR data confirming the proteomics “hits”. We have done such experiments for the main proteins on which we focused and for some others and we now include some of those data. They show (as expected) that some of the protein data conform with changes in mRNA levels and some do not – that is widely understood in the field and we now discuss that discrepancy in the text.

Therefore, it is clear that mRNA data (an imperfect measure of instantaneous rate of production) are not a reliable way to confirm the proteomics data (measures of steady-state protein levels). We also now include some confirmatory in vivo data from an autochthonous mouse model of mammary cancer. We believe that we have done the best that can be done – we have applied stringent criteria for scoring the proteomics data, all of which have been replicated. We have added text to the manuscript (pages noted in the response to referees, and highlighted in the resubmitted manuscript) to clarify for the reader. These are not one-off Western blot data and they are robust.

2) You also asked for secondary confirmation of “the role of these proteins in minimally another [xenograft] breast cancer cell line” and “additional knock down studies and characterization of this candidate protein and its role in a second breast cancer cell line” and you indicate that this “would be essential in order for the conclusions following from this work to be of broad interest to the readership of *eLife*.” While we understand the reasons for this request from you and the referees, we regret that we do not see it as a realistic one to achieve in the short term. As we have mentioned above, we did conduct RT-qPCR analyses on a variety of mammary cell lines but, as we demonstrate for the ones we have studied in detail (MDA-MB-MB231 and LM2), such data are not of much significance with respect to protein data. Furthermore, and perhaps even more important, in vitro data on other cell lines are very poorly relevant to the situation in vivo. All our data concern tumors growing in vivo and thus comprise both tumor-cell-derived proteins as well as stromal proteins and, as we show in the paper, there is extensive crosstalk between the cells – so in vitro data are not germane.

Thus, in order to do what you request, we would need to do in vivo confirmation on additional cell line tumors for protein expression, either by proteomics (an enormous amount of work) or, where antibodies are available (not the case for all the proteins of interest, although we are working on generating them) in order to validate changes in protein-level expression. Without such prior studies, we believe, it would be pointless to proceed to do further knockdown/metastasis assays (also a lot of time, mice, and expense) on other lines not validated for in vivo expression. Quite apart from the fact that what you are requesting would be a large body of work, requiring an extended time, large numbers of mice (we estimate that we used >100 mice for the experiments presented, each housed for months while tumors developed) and considerable expense both for the mice and for the proteomics, we do not believe that these are the next essential questions. If we believed that such a sequence of experiments would be productive, we would do them.

However, it is our opinion that they would not advance the situation. No matter how many cell lines or mouse models we examined, the question would still remain – how relevant are these xenotransplant data to human breast cancer? You and the referees have raised this point and we completely agree with it. Therefore, what we should be doing (and are) is extending the novel insights gained from this paper to human patient material. That is why we included some comparisons with patient outcome data extracted from the literature in our initial submission. We have now expanded that (and toned down some of our conclusions to “correlation” rather than “prediction”). These data show that many of the proteins we identify in vivo as upregulated in highly metastatic tumors show corroborative evidence – from other papers and/or from our comparisons with gene expression metadata analyses – supporting their roles in human breast cancer. Even though these are comparisons between protein level data and mRNA expression data with all their qualifications, there are a striking number of concordances that need investigation. We believe that validates our approach and the data presented in this paper – proteins we identify through in vivo analyses of our (admittedly) model discovery system include many that may be relevant to human cancer.

Therefore, we believe that the data we present in the revised manuscript do provide clear evidence that the proteins that we have discovered using a novel, systematic approach are clearly of interest for further work and to your readers. We see the next required steps to be to extend the work to human patient material, not to do further confirmatory experiments in xenotransplant models in mice. It has been our understanding that *eLife* also believes that excessive time should not be spent on performing experiments that do not materially improve a paper, delaying publication of the work, and postponing the progress of junior investigators onto the next substantive investigations. That is our contention here and we hope that you will consider our arguments for this position.